# Differing isoforms of the cobalamin binding photoreceptor AerR oppositely regulate photosystem expression

Haruki Yamamoto[†‡], Mingxu Fang[†§], Vladimira Dragnea, Carl E Bauer*

Department of Molecular and Cellular Biochemistry, Indiana University, Indiana, United States

**Abstract** Phototrophic microorganisms adjust photosystem synthesis in response to changes in light intensity and wavelength. A variety of different photoreceptors regulate this process. Purple photosynthetic bacteria synthesize a novel photoreceptor AerR that uses cobalamin ($B_{12}$) as a blue-light absorbing chromophore to control photosystem synthesis. AerR directly interacts with the redox responding transcription factor CrtJ, affecting CrtJ's interaction with photosystem promoters. In this study, we show that AerR is translated as two isoforms that differ by 41 amino acids at the amino terminus. The ratio of these isoforms was affected by light and cell growth phase with the long variant predominating during photosynthetic exponential growth and the short variant predominating in dark conditions and/or stationary phase. Pigmentation and transcriptomic analyses show that the short AerR variant represses, while long variant activates, photosynthesis genes. The long form of AerR also activates many genes involved in cellular metabolism and motility.

DOI: https://doi.org/10.7554/eLife.39028.001

*For correspondence:
bauer@indiana.edu

[†]These authors contributed equally to this work

Present address: [‡]Graduate school of Bioagricultural Sciences, Nagoya University, Nagoya, Japan; [§]Center for Circadian Biology, University of California, San Diego, United States

Competing interests: The authors declare that no competing interests exist.

## Introduction

Purple photosynthetic bacteria are an interesting group of diverse bacteria that preferentially synthesizes a photosystem under anaerobic conditions (*Imhoff and Hiraishi, 2015*). This is partially due to the fact that photopigments can generate singlet oxygen as a byproduct of light excitation (*Berghoff et al., 2011*; *Ziegelhoffer and Donohue, 2009*; *Pospísil, 2009*). Thus, many purple photosynthetic bacteria repress synthesis of their photosystem in response to the presence of oxygen, and in this case, generate energy using respiration (*Crofts et al., 1983*; *Swem et al., 2006*; *Wu and Bauer, 2010*). In species where it has been examined, there is an aerobic repressor called CrtJ (also called PpsR in some species) (*Elsen et al., 2005*; *Gomelsky and Kaplan, 1995*; *Penfold and Pemberton, 1991*; *Ponnampalam et al., 1995*) that senses oxygen via oxidation of a Cys present in CrtJ's DNA binding domain (*Cheng et al., 2012*; *Masuda and Bauer, 2002*; *Masuda et al., 2002*). Initially, it was thought that CrtJ/PpsR only binds and represses photosystem promoters under aerobic conditions (*Elsen et al., 2005*; *Ponnampalam et al., 1995*; *Cheng et al., 2012*; *Masuda and Bauer, 2002*; *Masuda et al., 2002*). However, a recent study showed that CrtJ is bound to a photosystem promoter (*bchC*) under both aerobic and anaerobic conditions (*Fang and Bauer, 2017*). Interestingly, the extent of DNA protection by CrtJ to this promoter is significantly altered in vivo under aerobic verses anaerobic conditions (*Fang and Bauer, 2017*).

In addition to environmental changes in oxygen tension, phototrophic microorganisms are subjected to daily variations in light intensity and wavelength. To optimize light energy absorption for photosynthesis, and to better adapt their physiology to dark periods, these cells also control photosystem synthesis using various photoreceptors (*Gomelsky and Hoff, 2011*; *Masuda, 2013*). A variety of chromophores such as FMN (used in Light, Oxygen or Voltage, LOV domain containing

**eLife digest** Some bacteria are able to use a process called photosynthesis to convert energy from sunlight into another form of energy they can use to grow. Within the bacteria, structures known as photosystems are responsible for absorbing light and transferring the energy to other molecules. The levels of light surrounding the bacteria continually fluctuate. To optimize the amount of light they absorb for photosynthesis, the bacteria have receptors that detect light and regulate the activities of the genes that produce photosystems.

One group of bacteria that carry out photosynthesis are collectively known as purple bacteria. These bacteria contain a light receptor called AerR that interacts with a protein called CrtJ, which can directly bind to and alter the activity of genes involved in photosynthesis. AerR senses light by binding to a molecule called vitamin B12, which can absorb blue light, but it was not clear how it affects the CrtJ protein.

Fang, Yamamoto et al. used biochemical and genetic approaches to study AerR in a purple bacterium known as *Rhodobacter capsulatus*. The experiments show that *R. capsulatus* makes two different versions of AerR. The larger version only binds to vitamin B12 that is carrying light energy and stimulates CrtJ to activate genes involved in photosynthesis. On the other hand, the shorter version binds to vitamin B12 in the dark and causes CrtJ to repress genes that produce photosystems.

Receptors similar to AerR are found in many bacteria and other single-celled organisms known as Archaea, including in species that do not perform photosynthesis. Therefore, these findings are likely to be useful to researchers studying how bacteria and Archaea sense light in a variety of situations. A next step will be to find out how the different forms of AerR can change the properties of CrtJ.

DOI: https://doi.org/10.7554/eLife.39028.002

proteins), FAD (used in sensors of Blue-Light Using FAD, BLUF domain containing proteins), 4-hydroxycinnamic acid (used in Photoactive Yellow Protein, PYP), phytochromobilin (used in bacterio-phytochromes) and cobalamin ($B_{12}$) are used as light absorbing chromophores by photoreceptors present in purple bacteria (*Masuda and Bauer, 2002*; *Gomelsky and Hoff, 2011*; *Masuda, 2013*; *Gomelsky and Klug, 2002*; *Hendriks, 2009*; *Giraud et al., 2002*; *Jiang et al., 1999*; *Cheng et al., 2014*; *Cheng et al., 2016*). Photoreceptor proteins need to either contain a linked output domain or to interact with other proteins that transmit photoreceptor sensed light signals into physiological changes. In some purple bacteria, photoreceptors can interact with transcription factors to alter the transcription of genes (*Hendriks, 2009*). For example, in the purple non-sulfur alpha-proteobacterium *Rhodobacter sphaeroides*, the interaction of the FAD-binding blue light sensor AppA with the transcription factor PpsR is well studied (*Masuda and Bauer, 2002*; *Giraud et al., 2002*; *Gomelsky and Kaplan, 1997*; *Metz et al., 2012*). In *Rhodopseudomonas palutris*, PpsR repression is also regulated via an interaction with a red-light sensing bacteriophytochrome (*Braatsch et al., 2006*; *Braatsch et al., 2007*). These light dependent interactions have also been used as optogenetic tools (*Kaberniuk et al., 2016*; *Redchuk et al., 2017*).

In all sequenced purple photosynthetic bacterial genomes, there is a gene coding for a cobalamin binding photoreceptor called *aerR* that is located upstream of *ppsR/crtJ* (*Cheng et al., 2014*; *Vermeulen and Bauer, 2015*). The discovery that AerR binds cobalamin in a light-dependent manner explained a long-standing observation that this group of bacteria needs cobalamin to undergo synthesis of their photosystem (*Cheng et al., 2014*). However, it remains unknown how AerR and CrtJ coordinate global changes in *R. capsulatus* physiology in response to changes in redox and light. We have shown that AerR can directly interact with CrtJ both in vivo and in vitro (*Fang and Bauer, 2017*) and that AerR can affect the DNA binding characteristics of CrtJ at target promoters (*Fang and Bauer, 2017*). In this study, we demonstrate that *R. capsulatus* uses two offset promoters, and an alternative internal Leu start codon, to synthesize long and short isoforms of AerR. We further show that the long AerR variant converts CrtJ into an activator while the short AerR variant converts CrtJ into a repressor of photosystem gene expression. We also show that the long AerR variant

coordinates synthesis of photosystem with expression of a large variety of additional genes that affect cellular metabolism and motility.

## Results

### There are two isoforms of AerR in vivo

We analyzed the in vivo presence of AerR by Western blot analysis using an *R. capsulatus* strain in which a 3xFLAG-tag was chromosomally inserted at the 3′ terminus of the *aerR* open reading frame. A previous study established that a 3xFLAG-tag at the carboxyl end of AerR did not measurably affect AerR activity (*Cheng et al., 2014*). As indicated in *Figure 1B*, Western blot analysis shows the unexpected presence of two AerR-FLAG isoforms with one at ~30 kDa (LAerR) and the other at ~25 kDa (SAerR) based on electrophoretic mobility. The larger 30 kDa isoform present in *R. capsulatus* extracts co-migrates with full-length (based on mass-spectrophotometry analysis) AerR-FLAG protein that was expressed and purified from *Escherichia coli* (*Cheng et al., 2014*). This indicates that the 25 kDa isoform (SAerR) either represents a proteolytic processing event or an alternative translational start site. Interestingly, steady-state amounts of these two AerR isoforms consistently changed depending on what point in the *R. capsulatus* cultivation growth curve that AerR-FLAG isoforms were analyzed (*Figure 1A,B*). When dark semi-aerobically grown cells were harvested at different points in mid-exponential phase, these two AerR isoforms are present at approximately equal amounts. However, when harvested in stationary phase, the 25 kDa SAerR isoform was constantly the dominant form.

When grown under anaerobic photosynthetic (constant illumination) conditions, the large isoform predominates in early and mid-exponential phase and then decreases in concentration as the culture transitions to late exponential and stationary phases, leading to a predominance of the small isoform (*Figure 1—figure supplement 1*).

In addition to growth culture changes in the ratio of LAerR and SAerR, a change of the LAerR/SAerR ratio is also observed upon a light shift. For example, switching dark exponentially grown cells at an optical density of 0.3 to anaerobic illuminated conditions, results in a significant increase in LAerR relative to a parallel fraction of cells that remained in the dark (*Figure 1C*). Conversely, switching anaerobic photosynthetically cells from light to dark conditions, reduced the level of LAerR (*Figure 1C*).

### Transcriptional and translational properties of the *aerR* gene

The difference in electrophoretic mobility between LAerR and SAerR indicates that there are ~40–50 fewer amino acid residues in the N-terminal region of the SAerR isoform. We first addressed whether SAerR is a product of proteolytic truncation of LAerR by constructing an LAerR frameshift mutation via the insertion of one nucleotide immediately downstream of the LAerR Met initiation codon (Met1a) (*Figure 2A*). The resultant *aerR*-Met1a construct, containing a carboxyl FLAG tag sequence, was then ectopically expressed on a plasmid using its own native promoter. Western blot analysis demonstrated that the *aerR*-Met1a plasmid only expressed the 25 kDa SAerR isoform (*Figure 2A*). This indicates that the small isoform is not derived by proteolytic processing of LAerR, and must be generated from a second internal translation initiation site present in the LAerR coding sequence.

There are several additional in-frame Met codons in the 5′ *aerR* coding sequence (Met35, Met49 and Met61) that could potentially function as a second initiation codon for SAerR. We accessed the possibility that one of these internal Met codons was functioning as an internal SAerR initiation codon by constructing three deletions within the LAerR coding region (*Figure 2A*). The first deletion extended from the LAerR initiation codon to Met35, the second deletion extended from the LAerR initiation codon to Met49 and the third deletion extended from the LAerR initiation codon to Met61. Each of these AerR deletion constructs also contained a C-terminal FLAG epitope and all were ectopically expressed from the *aerR* native promoter in *R. capsulatus* on a plasmid. As shown in *Figure 2A*, the Met1 through Met35 truncated strain still expressed two forms with the 30 kDa isoform shifted to ~27 kDa as a result of the deletion/truncation while the 25 kDa isoform showed same mobility as the WT control. The second deletion from Met1 through Met49 and the third deletion that extended to Met61 lost expression of both the longer and shorter isoforms indicating that the SAerR isoform starts between the M35 and M49 codons. One nucleotide frame shift insertions

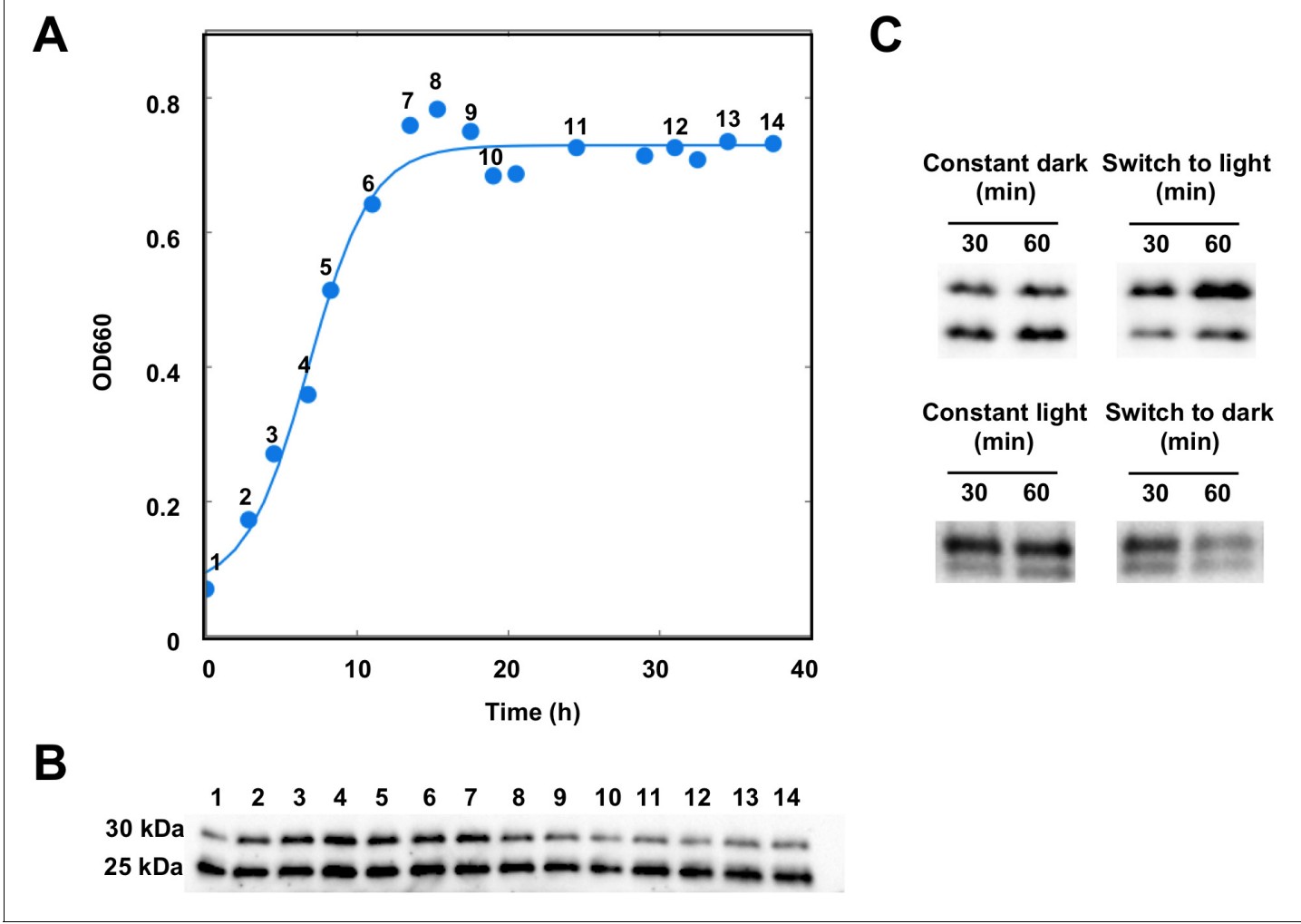

**Figure 1.** AerR exists as two forms depending on the position in a cells growth cycle. (A) *R. capsulatus* grown under aerobic condition and AerR-FLAG protein expression was checked at each point (1 to 14 in growth curve). (B) Western blot analysis to detect AerR-FLAG protein. Lane numbers are responding numbers in the growth curve. (C) AerR expression was checked when cells were shifted from dark to light or from light to dark after 30 and 60 min.

DOI: https://doi.org/10.7554/eLife.39028.003

The following figure supplements are available for figure 1:

**Figure supplement 1.** AerR also exists as two forms under photosynthetic growth conditions.

DOI: https://doi.org/10.7554/eLife.39028.004

**Figure supplement 2.** AerR (PpaA) expression pattern in other species.

DOI: https://doi.org/10.7554/eLife.39028.005

were subsequently created at several points between the Met35 and Met49 codons to further determine the SAerR initiation site (*Figure 2B*). While M35$^{+1}$ (one nucleotide inserted after M35 codon) and V38$^{+1}$ (one nucleotide inserted after V38 codon) strains still expressed the 25 kDa isoform, strains with nucleotide insertions after the L41, T43, and V44 codons did not express SAerR (*Figure 2B*). This result indicates the second initiation starts at the A39, E40, or L41 codon.

To find the exact location of the SAerR initiation codon, we constructed plasmids that contained the *aerR* promoter region and a partial *aerR* sequence fused to the *bchE* open reading frame that also had a FLAG epitope tag for use as a reporter (*Figure 2C*). To focus on identification of the second initiation codon, these constructs also contained a one base insertion downstream of the LAerR initiation codon, resulting in disruption of LAerR peptide synthesis. Three constructs were made; one having the FLAG-*bchE* gene fused downstream of the A39 *aerR* codon, a second construct with

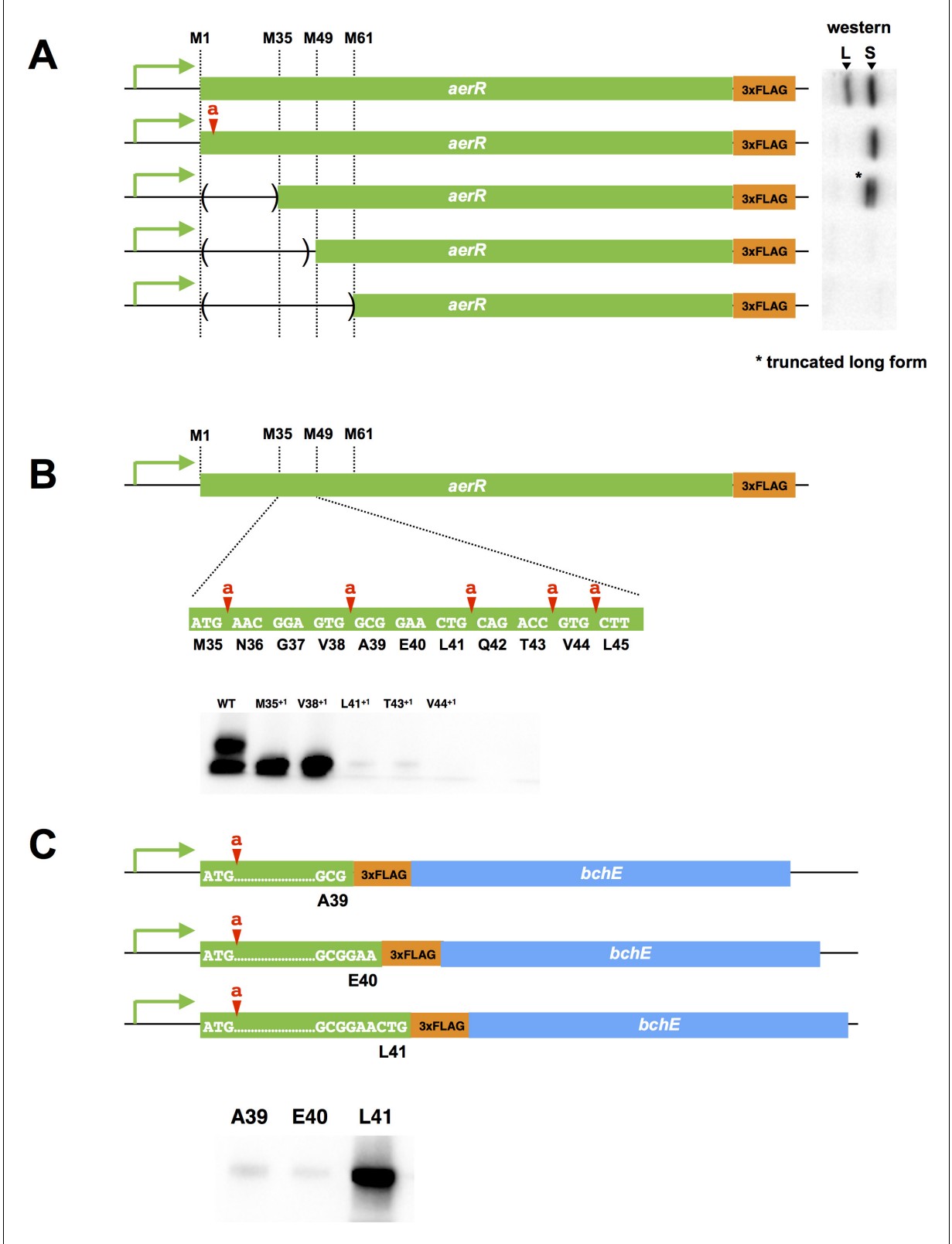

**Figure 2.** Mutational analysis of the internal second initiation codon, Leu41 (CTG). All AerR mutant proteins were detected by Western blot analysis with anti-FLAG antibody. (A) Several truncated AerR proteins were expressed in *R. capsulatus* from internal start codon candidates, M35, M49, and M61, respectively. Frameshift mutations were introduced via one nucleotide insertions downstream of the Met1 codon. (B) One nucleotide was inserted after M35, V38, L41, T43, and V44 codons, respectively. (C) Three partial *aerR* sequences, that include the M1 codon to the A39, E40, or L41 codons,

*Figure 2 continued on next page*

*Figure 2 continued*

were fused to the *bchE* open reading frame that also had a FLAG epitope tag as a reporter. Translation from the M1 codon was blocked by a one nucleotide insertion after the M1 codon.

DOI: https://doi.org/10.7554/eLife.39028.006

The following figure supplement is available for figure 2:

**Figure supplement 1.** Summary of AerR translation start sites and AerR mutant construction.

DOI: https://doi.org/10.7554/eLife.39028.007

FLAG-*bchE* fused to *aerR* codon E40, and a third with FLAG-*bchE* fused to L41. When these *aerR-bchE*-FLAG reporter plasmids were introduced into *R. capsulatus,* only the plasmid that had *bchE*-FLAG fused to L41 resulted in expression of a BchE-FLAG protein (*Figure 2C*). This indicates that the L41 codon (CTG) is functioning as a second start site for the SAerR isoform. Six through sixteen bases upstream of L41 is a putative ribosome binding site (5'-GAAcGGAGtgG-3') that exhibits considerable complementarity with the cognate *R. capsulatus* 16S rRNA sequence (3'-CUUuCCUCcaC-5').

Finally, we also analyzed whether there was an additional internal transcription start site for SAerR by amplification of 5' mRNA end(s) using cDNA 5' RACE (Rapid Amplification of cDNA Ends) (GeneRacer kit, Invitrogen). Note that a terminator exonuclease (Epicentre) treatment step was introduced in the 5' RACE experiments to degrade processed RNA, allowing only the selection for primary transcripts. A 24 base cDNA primer was also designed to be complementary to a region of the *aerR* transcript that is 299 bases downstream of the LAerR initiation codon. Consequently, this primer should amplify both a large transcript responsible for transcribing LAerR as well as a shorter transcript responsible for transcribing SAerR, should such a shorter transcript exist. As shown in *Figure 2—figure supplement 1A*, two transcription start sites were indeed detected by sequencing: one with a start site 25 bp upstream of the LAerR initiation codon and a second initiating at the 8th codon of LAerR (*Figure 2—figure supplement 1A*). The longer transcript could potentially be responsible for both LAerR and SAerR expression while the shorter transcript would be dedicated to the expression of SAerR. Inspection of the sequence upstream of these transcript initiation sites shows the presence of putative promoter recognition sequences with the SAerR −35 and −10 regions exhibiting good sequence similarity to previously characterized *R. capsulatus* promoters (*Swem et al., 2001*). However, this is not the case with the upstream LAerR promoter region which shows poor promoter sequence conservation indicating that the upstream LAerR promoter may utilize an alternative sigma subunit (*Figure 2—figure supplement 1A*). Furthermore, there are FnrL binding sites located near both promoter regions, which is not unexpected given that FnrL is known to regulate AerR expression (*Kumka and Bauer, 2015*) and that ChIP-seq analysis also has revealed FnrL binds at both of these promoter regions in vivo (*Kumka and Bauer, 2015*).

## Construction of strains that express stable AerR isoforms

To effectively evaluate the function of each AerR isoform required that strains be constructed that only express the long or short isoforms of AerR. To disrupt SAerR synthesis, we constructed a single silent mutation of L41 (Leu CTG to Leu CTC) as well as a second silent mutation that also disrupted the upstream ribosome binding site (RBS) (*Figure 2—figure supplement 1B*). The RBS silent mutation changed the Glu37 codon from GGA to an alternate Glu codon GGT resulting in conversion of the GAAcGG**A**GtgG ribosome recognition sequence to GAAcGG**t**GtgG. The combination of both of these silent mutations resulted in a strain (termed ΔSAerR) that produces normal amounts of LAerR without any detectable amounts of SAerR (*Figure 2—figure supplement 1C*).

To obtain a strain that only synthesizes SAerR, we constructed a strain that had a one nucleotide chromosomal insertion immediately downstream of the LAerR Met initiation codon (*Figure 2—figure supplement 1B*). This frameshift insertion results in the generation of an in frame stop codon 15 bp downstream of the LAerR initiating Met codon resulting in the synthesis of just a five aa peptide. Western blot analysis shows that this strain (termed ΔLAerR) only synthesizes SAerR (*Figure 2—figure supplement 1C*). Finally, a negative control strain was constructed (AerR null) that lacks both the short and long forms of AerR. This strain has a frameshift after codon L45 that results in undetectable amounts of both the LAerR and SAerR peptides (*Figure 2—figure supplement 1C*). All of these

constructions were recombined into the *R. capsulatus* chromosome at their native location under control of the described AerR promoters.

## Divergent photopigment phenotypes are obtained by strains expressing different isoforms of AerR

We measured the in vivo production of photopigments on strains containing a deletion of either the long or short AerR isoforms, relative to wild-type and the *aerR* null mutant strains. For this analysis, cells were grown in rich PY medium under dark semi-aerobic conditions, harvested at late-exponential phase (OD = 0.6 to 0.7) and analyzed for the level of pigments after organic extraction. The bar graph in *Figure 3A* shows that relative to wild-type cells, the strain that lacks the short isoform (ΔSAerR) has ~1.6 and 1.75-fold enhancement of carotenoid and bacteriochlorophyll (Bchl) levels, respectively. This is contrasted by the strain which lacks the long isoform of AerR (ΔLAerR) that exhibits significantly reduced amounts of carotenoids and Bchl (20% and 2% of WT, respectively). The amount of Bchl biosynthesis exhibited by the strain lacking LAerR is also lower than that observed with the AerR null strain (20% of WT), which is thought to undergo constitutive CrtJ mediated repression of *bch* gene expression (*Fang and Bauer, 2017*). In this growth condition, it appears that the long and short isoforms of AerR have opposite functions with SAerR involved in suppressing photopigment synthesis and LAerR involved in enhancing photopigment biosynthesis. When these same four strains were grown under anaerobic photosynthetic illuminated conditions, the ΔSAerR strain exhibited a slight 30% reduction in Bchl and carotenoid production while the ΔLAerR strain showed a more severe reduction in these pigments (78% and 65% reduction, respectively) relative to WT cells (*Figure 3B*). Photosynthetic pigment reduction observed upon an absence of LAerR is slightly lower than observed with the AerR null strain that had 72% and 63% reduction in Bchl and carotenoids relative to WT cells.

When shifted from aerobic to anaerobic photosynthetic growth conditions, the WT and ΔSAerR strains both exhibited an 8 hr lag followed by comparable growth rates (*Figure 3C*). In contrast to good photosynthetic growth by the ΔSAerR strain, the AerR null and ΔLAerR strains both exhibited a more severe lag upon the shift to photosynthetic growth conditions (~30 and 60 hr, respectively) as compared to the WT strain (*Figure 3C*). This lag in growth is presumably a result of decreased pigment biosynthesis by the ΔLAerR and AerR null strains.

## CrtJ dependency of the AerR phenotype

Studies have shown that AerR alone does not appear to contain the ability to directly interact with DNA (*Fang and Bauer, 2017*; *Cheng et al., 2014*). However, these studies have also demonstrated that AerR does tightly interact with the photosystem regulator CrtJ in vivo and in vitro. This interaction also subsequently affects the DNA binding characteristics of CrtJ (*Fang and Bauer, 2017*; *Cheng et al., 2014*). To further understand the role of short and long forms of AerR, we checked whether both isoforms are indeed both capable of interacting with CrtJ. We also checked the dependency of CrtJ on the observed phenotypes of strains expressing only SAerR or LAerR.

In vitro binding affinities measuring the interaction of each AerR isoform with CrtJ were obtained using microscale thermophoresis (MST) as previously reported (*Cheng et al., 2014*). The observed binding affinity of CrtJ to LAerR was slightly lower than observed with CrtJ to SAerR with an $EC_{50}$ = 7.8 ± 1.2 µM for LAerR containing bound hydroxyl-Cbl (OH-Cbl), 2.1 ± 1 µM for SAerR with bound OH-Cbl and 2.1 ± 0.8 µM SAerR with bound Adenosyl-Cbl (Ado-Cbl). Thus, both AerR isoforms are indeed capable of forming a complex with CrtJ. The type of Cbl bound to SAerR also does not seem to appear affect its interaction with CrtJ.

We next addressed whether LAerR and SAerR both interact with CrtJ in vivo by addressing CrtJ dependency on the observed phenotypes exhibited by strains that lacked either LAerR or SAerR. A 1.5-fold increase in dark semi-aerobic Bchl production has previously been reported for the *crtJ* deletion strain (*Cheng et al., 2012*) so the question we addressed is whether increased Bchl production exhibited by a CrtJ deletion mutation is dominant over the observed reduction in Bchl production by the ΔLAerR and AerR null strains when grown under dark semi-aerobic conditions. For this analysis, we constructed an in-frame deletion of *crtJ* in the relevant *aerR* mutant strains giving rise to Δ*SaerR*-Δ*crtJ*, Δ*LaerR*-Δ*crtJ* and null *aerR*-Δ*crtJ* strains. In dark semi-aerobic growth conditions, all three strains that contained the *crtJ* deletion (Δ*crtJ*, null *aerR*-Δ*crtJ*, Δ*LaerR*-Δ*crtJ* and Δ*SaerR*-Δ*crtJ*)

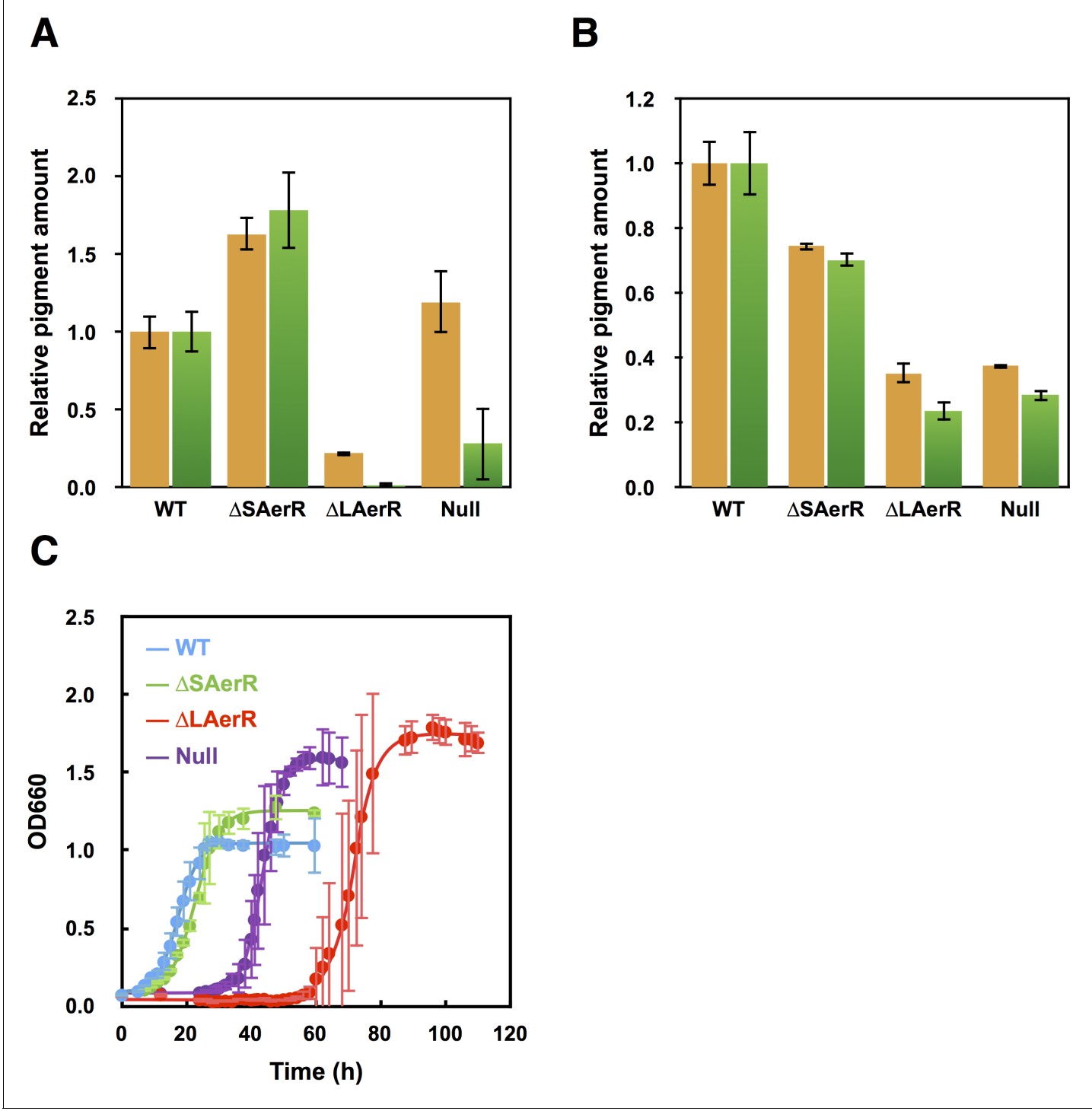

**Figure 3.** Pigment levels in WT, ΔSAerR, ΔLAerR and negative control null AerR strains and their effect on photosynthetic growth. Total pigment obtained from organic extraction from (**A**) aerobically or (**B**) photosynthetically grown WT, ΔSAerR, ΔLAerR or negative control null AerR strains that were harvested at late-exponential phase (OD = 0.6 to 0.7). Yellow bars indicate carotenoids and green bars indicate bacteriochlorophyll. (**C**) Photosynthetic growth of WT (blue), ΔSAerR strain (green), ΔLAerR strain (red), and the AerR null strain (purple) after shifting from dark semi-aerobic to photosynthetic illuminated conditions.

DOI: https://doi.org/10.7554/eLife.39028.008

The following figure supplement is available for figure 3:

**Figure supplement 1.** SAerR and LAerR phenotypes are suppressed by a CrtJ null mutation.

*Figure 3 continued on next page*

*Figure 3 continued*

DOI: https://doi.org/10.7554/eLife.39028.009

exhibited 1.5 to 2-fold higher amounts of Bchl relative to that observed by the WT control (*Figure 3—figure supplement 1A*). Interestingly, this increase in pigment production is very similar to the increase in pigment production observed when the ΔSAerR strain just expresses the LAerR isoform (*Figure 3A*). The 1.5-fold increase in Bchl production over that of WT strain exhibited by the ΔL*aerR*-Δ*crtJ* strain (*Figure 3—figure supplement 1*) is also a stark contrast to the severe ~80% reduction in pigment production exhibited by the ΔLAerR strain (*Figure 3A*). This result demonstrates that the increased pigment production exhibited by a deletion of CrtJ is dominant over reduced pigment production exhibited by the loss of LAerR. Furthermore, the observed increased pigment production exhibited by the ΔSAerR strain appears to be indistinguishable from the increased pigmentation phenotype exhibited by a deletion of CrtJ. One conclusion in comparing the results in *Figure 3* and in *Figure 3—figure supplement 1* is that (i) both LAerR and SAerR works in CrtJ dependent manner and (ii) an absence of SAerR results in CrtJ no longer being able to repress bacteriochlorophyll gene expression whereas a loss of the LAerR isoform leads to the inability of CrtJ to enhanced pigment production.

When pigment production was analyzed under anaerobic photosynthetic conditions, we observed that the introduction of the Δ*crtJ* deletion into the AerR null, ΔSAerR and ΔLAerR strains largely suppresses these AerR mutant phenotypes (*Figure 3—figure supplement 1B*) indicating that the Δ*crtJ* phenotype is dominant over that of the ΔSAerR and ΔLAerR phenotypes under photosynthetic conditions. Finally, the severe lag in photosynthetic growth exhibited by the ΔLAerR strain and the null AerR strains (*Figure 3C*) was also suppressed by introduction of a Δ*crtJ* deletion (*Figure 3—figure supplement 1C*).

## SAerR exhibits altered cobalamin binding activity

A previous study showed that LAerR binds cobalamin (Cbl) in a light-dependent manner (*Cheng et al., 2014*). Specifically, it was shown that LAerR tightly binds OH-Cbl which is generated as a byproduct of light excitation of Ado-Cbl (*Cheng et al., 2014*). A Cbl deficiency was also shown to result in a reduction in pigmentation and photosystem gene expression in a manner that mimics the phenotype of an AerR null mutation. Furthermore, in vitro studies have shown that LAerR forms upper and lower axial ligands with the Co metal in OH-Cbl using two histidine residues (His10 and His145). Alanine substitutions on one of these His ligands alters the Co spectrum with Ala mutations in both of these His residues abolishing the ability of LAerR to bind OH-Cbl (*Cheng et al., 2014*). Given that SAerR lacks the His10 upper Co ligand, we also addressed whether SAerR is indeed capable of light-dependent binding OH-Cbl in vitro as described for LAerR. For this analysis, purified SAerR protein was incubated with several Cbl derivatives under dark or illuminated conditions followed by removal of unbound cobalamins with a desalting column (*Figure 4A*). Spectral analysis of SAerR surprisingly showed that SAerR is able to bind all tested cobalamin derivatives (OH-Cbl, Ado-Cbl, cyano-Cbl, and methyl-Cbl) under both dark and illuminated conditions (*Figure 4B* and *Figure 4—figure supplement 1*). This result indicates that His10 in the amino-terminal region of LAerR appears to be responsible for the selectivity of light generated OH-Cbl.

An SAerR construct incapable of binding Cbl was also constructed to evaluate Cbl dependency on SAerR activity. For this analysis, we constructed an Ala mutation at H145 in the cobalamins binding motif ($E_{143}xH_{145}xxG_{148}$) that in LAerR is known to form a lower axial ligand with the Cbl bound Co (*Cheng et al., 2014*). As shown in *Figure 4C*, the H145A substitution largely disrupted SAerR ability to bind Cbl. To completely disrupt Cbl binding we also constructed a second mutation in Gly148 in the cobalamins binding motif to Glu. The SAerR_H145A, G148E double mutant protein no longer bound any detectable amounts of Cbl (*Figure 4C*).

To evaluate the Cbl dependency on SAerR repressor activity in vivo, this SAerR double mutant protein was also expressed in *R. capsulatus* cells. As shown in *Figure 5*, a WT strain harboring an SAerR expression plasmid exhibited much less Bchl and carotenoid synthesis than did a WT strain without the SAerR expression plasmid (red versus blue spectrum in *Figure 5A*, respectively). This pigment reduction mirrors the results in *Figure 3* which show that a strain that just harbors SAerR

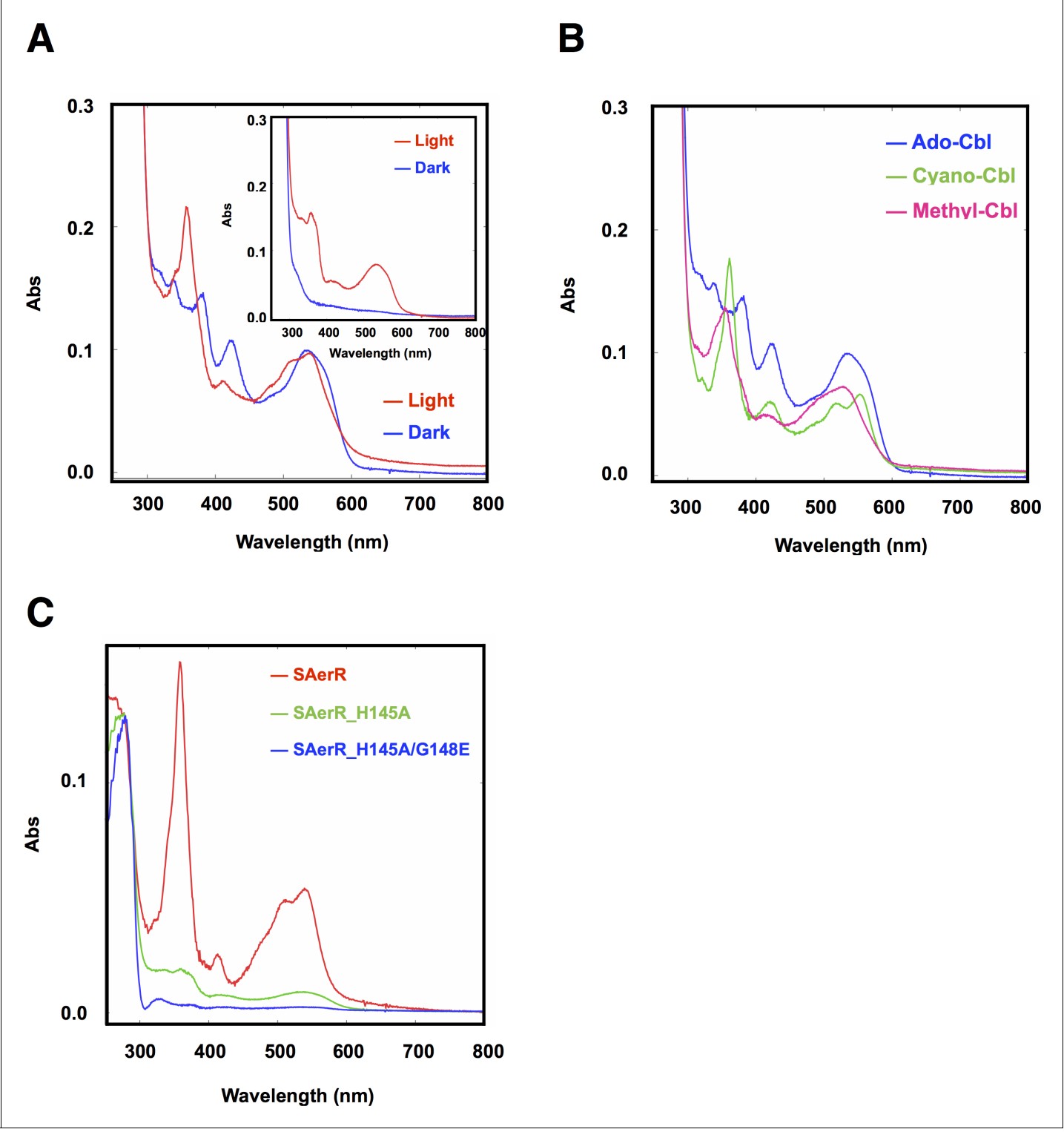

**Figure 4.** Cobalamin binding ability of SAerR protein. (**A**) The insert shows a spectral scan of purified LAerR protein incubated with Ado-Cbl in dark (blue) or light (red) for 5 min followed by removal of unbound cobalamins. LAerR only binds OH-Cbl which is a product of light excitation of Ado-Cbl, thus LAerR does not bind Cbl under dark conditions. The larger spectrum is of SAerR incubated with Ado-Cbl in dark (blue) or light (red) for 5 min followed by removal of unbound cobalamins. SAerR binds both Ado-Cbl in the dark as well as light generated OH-Cbl. (**B**) Spectral scan of SAerR incubated with Ado-Cbl (blue), Cyano-Cbl (green), or Methyl-Cbl (pink) in dark for 5 min followed by removal of unbound cobalamins. (**C**) Spectral scan of purified SAerR (red), SAerR_H145A (green), and SAerR_H145A/G148E (blue) incubated with Ado-Cbl in light illumination for 5 min followed by removal of unbound cobalamins.

*Figure 4 continued on next page*

*Figure 4 continued*

DOI: https://doi.org/10.7554/eLife.39028.010

The following figure supplement is available for figure 4:

**Figure supplement 1.** SAerR binds several kinds of cobalamin.

DOI: https://doi.org/10.7554/eLife.39028.011

has a severe reduction in pigment synthesis. In contrast, a strain harboring an SAerR_H145A,G148E expression plasmid contained only slightly less than WT amounts of Bchl and carotenoid (orange spectrum in *Figure 5A*). This indicates that disruption of SAerR's ability to bind $B_{12}$ impairs CrtJ mediated repression activity. The growth rate observed when shifted from semi-aerobic to anaerobic photosynthetic conditions also supports this conclusion (*Figure 5B*). Specifically, the SAerR_H145A, G148E expressing strain grew at nearly the same growth rate as the strain that did not express SAerR while the strain that expressed SAerR exhibited a significant delay in photosynthetic growth. Collectively, these results indicate that unlike LAerR, SAerR promiscuously binds Cbl with differing upper ligands in a light-independent manner and that Cbl binding has an important role for SAerR activity. That said, we do note that the Cbl binding mutant does not reach the same optical density as does the WT strain, and also has pigment levels that do not reach the same level as that of the WT strain. This suggests that apo-SAerR may also have a yet undefined role in these cells.

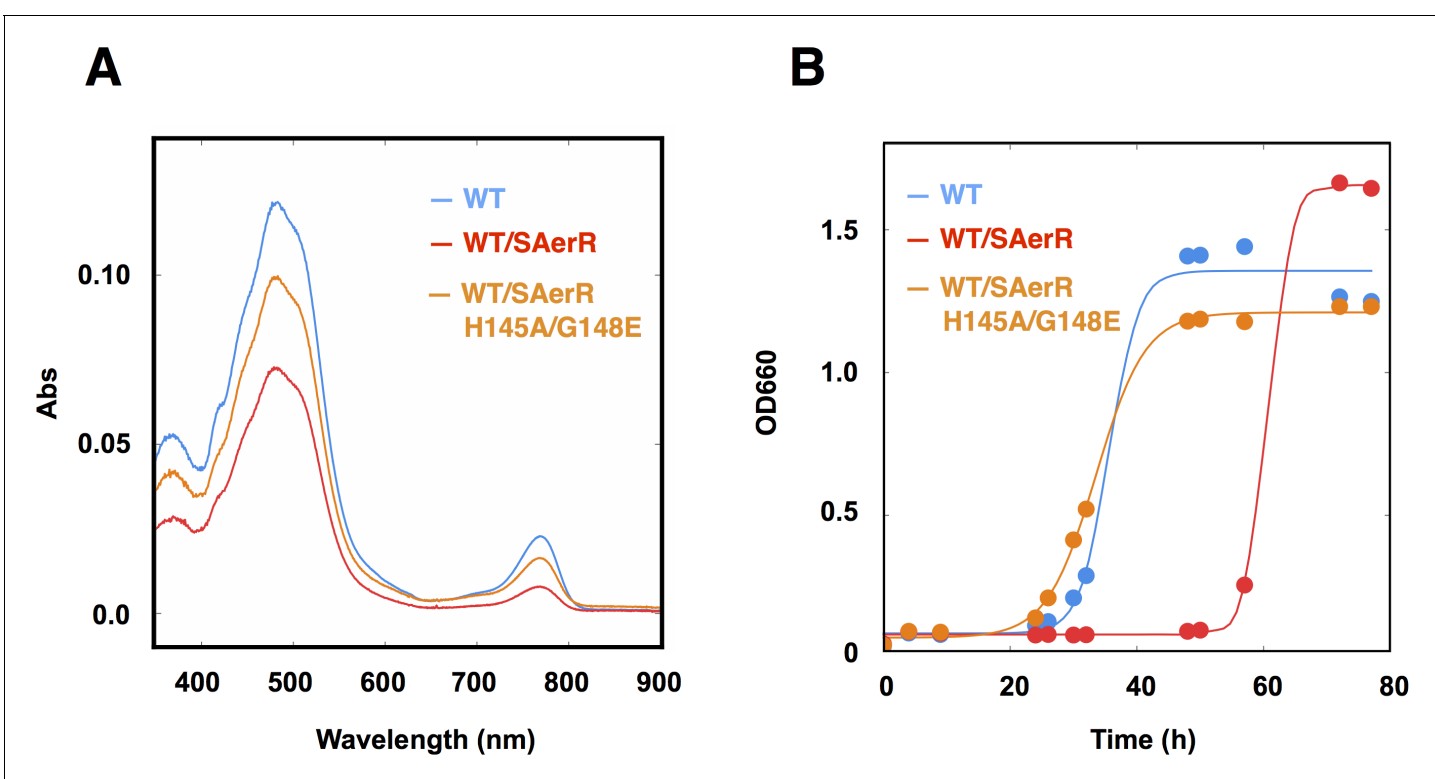

**Figure 5.** An SAerR overexpression phenotype is suppressed in vivo by mutations in the cobalamin binding motif. (A) Spectrum of total pigment extracts from semi-aerobically grown WT strain (blue), a WT strain overexpressing SAerR (red), and a WT strain overexpressing SAerR_H145A/G148E (orange). (B) Growth curve of WT cells (blue), WT cells overexpressing SAerR (red), and WT cells overexpressing SAerR_H145A/G148E (orange). These cells were shifted from dark semi-aerobic to photosynthetic growth conditions.

DOI: https://doi.org/10.7554/eLife.39028.012

## Photosynthesis gene expression is differentially regulated by LAerR and SAerR

We next explored changes in gene expression patterns in the strains lacking LAerR or SAerR by differentially comparing their transcriptomes with the transcriptome of the WT strain using RNA-seq. The heat map in *Figure 6A*, and the quantitated fold-changes in *Supplementary file 1*, show photosynthesis gene expression changes under dark semi-aerobic conditions. Overall the ΔLAerR strain that lacks LAerR, and the AerR null mutation strain that does not contain either isoforms of AerR, both exhibited reduced photosystem gene expression profiles relative to that observed by the WT strain. However, there are several important differences. For example, the ΔLAerR strain has significantly reduced expression of the *puc* operon coding for light harvesting II (LHII) structural peptides which is not observed by the AerR null strain where *puc* expression is unchanged from the WT strain (*Figure 6A*). This result indicates that SAerR has a role in repressing *puc* (LHII) expression. A second

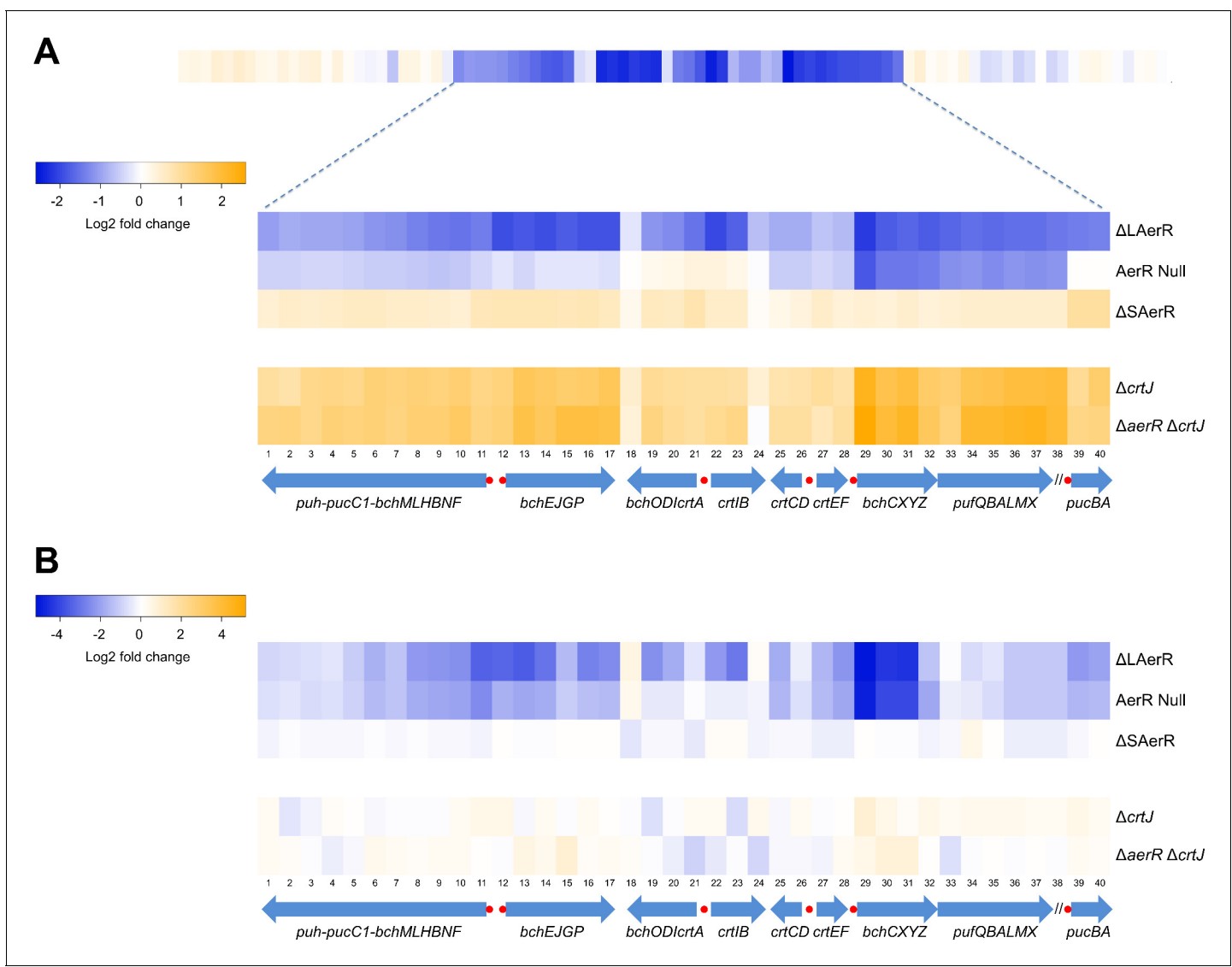

**Figure 6.** Heat map of global expression changes in AerR and CrtJ mutant strains as measured by RNA-seq. (**A**) Changes in expression occuring in the photosynthesis gene cluster region of the *R. capsulatus* chromosome in dark semi-aerobically grown AerR mutation strains. The ΔLAerR and AerR null strains exhibit severe reduction in photosystem transcription while the ΔSAerR exhibits an increase in expression. The reduced expression phenotypes exhibited by the ΔLAerR and AerR null strains are overcome by introduction of a *crtJ* deletion. (**B**) Similar to A with the exception that the cells were grown anaerobically under photosynthetic illuminated conditions. CrtJ binding motifs are shown as a red-dots.
DOI: https://doi.org/10.7554/eLife.39028.013

difference is that the ΔLAerR strain also has significantly reduced expression of the *bchEJGP* operon and reduced expression of the divergent *crtA-bchIDO* and *crtIB* operons relative to the AerR null and WT strains (*Figure 6A*, *Supplementary file 1*). This also indicates a role of SAerR in repressing expression of these operons. Note that the *crtI-crtB* gene products code for phytoene dehydrogenase and phytoene synthase, respectively, which are involved in the first two committed steps of carotenoid biosynthesis (*Armstrong, 1995*). The same is true for *bchID* which code for the ATPase subunits in Mg-chelatase, the first committed step in Bchl biosynthesis (*Senge and Smith, 1995*). Thus, the rather significant 'super' repression of these two divergent operons that occurs upon loss of LAerR likely causes the observed severe reduction of Bchl and carotenoids synthesis in the dark semi-aerobically grown ΔLAerR strain (*Figure 3A*). Evidence for the involvement of CrtJ in LAerR mediated activation of these photosystem genes is also observed by the rather significant increase in expression exhibited by the Δ*crtJ* and Δ*aerR*Δ*crtJ* double mutant strains (*Figure 6A*). This is also congruent with the observed photopigment phenotype suppression in the SAerR and LAerR strains by the addition of a CrtJ mutation as seen in *Figure 3—figure supplement 1*. Finally, in regards to the ΔSAerR strain that lacks SAerR, there is increased photosystem transcripts relative to WT cells indicating that SAerR likely has a repressing role under dark semi-aerobic growth conditions (*Figure 6A*).

When analyzing the transcriptome under illuminated anaerobic photosynthetic conditions (*Figure 6B*, *Supplementary file 2*), the transcriptome profiles again largely mimic the pigment levels observed by these strains (*Figure 3B*). Specifically, the ΔLAerR strain that lacks LAerR shows a reduced photosystem expression profile relative to WT cells indicating that LAerR also has an important role in activating photosystem gene expression anaerobically. Again, this is particularly evident for the *bchEJGP* operon and the divergent *crtA-bchIDO* and *crtIB* operons. In contrast, the ΔSAerR strain lacking SAerR has only a minor reduction in photosystem gene expression relative to WT cells indicating that SAerR has a minor, or even no role, in controlling photosystem gene expression under anaerobic photosynthetic growth conditions.

## LAerR controls gene expression well beyond that of photosystem genes

Additional analysis of RNA-seq results from the AerR null strain under dark semi-aerobic growth conditions revealed that this strain had only eight genes, in addition to *bch, crt* and photosystem structural genes (*puf, puc, puh*), that exhibited significant differential expression relative to WT cells (*Supplementary file 3*, tab 1). This indicates that the primary role of AerR under dark semi-aerobic growth conditions, a condition where SAerR predominates, is to control the expression of photosynthesis genes. However, the limited dark semi-aerobic regulatory role is contrasted by analysis under illuminated anaerobic photosynthetic conditions where deletion of AerR affects the expression of >1500 genes (*Supplementary file 3*, tab 2). In this growth mode LAerR predominates and seems to have a significant role in controlling cellular physiology well beyond that of photosynthesis.

We specifically addressed the involvement of the LAerR isoform in controlling global cellular physiology under photosynthetic conditions by analyzing the RNA-seq transcriptome profile in the ΔLAerR strain. As shown in *Supplementary file 4* and *Figure 7*, the AerR null strain that lacks both isoforms and the ΔLAerR strain that lacks only LAerR, both exhibited differential expression changes (relative to the WT strain) that were very similar to each other. Interestingly, in the few cases where loss of SAerR has an effect on gene expression under photosynthetic conditions, there is an inverse effect relative to that observed upon loss of LAerR. Thus, in cases where LAerR functions as an activator, SAerR appears to function as a repressor and vice versa (*Figure 7*, *Supplementary file 4*).

When assessing the role of individual genes that are regulated by LAerR photosynthetically (*Supplementary file 4*, tab subcategories and summarized in *Figure 8*), we observed the following. A loss of the LAerR isoform results in lower expression of genes involved in such diverse cellular processes as photosynthesis, carbon fixation, chemotaxis and motility, cobalamin biosynthesis, glycolysis and TCA cycle, heme biosynthesis, ribosomal proteins and several transporters. In each these cases, LAerR appears to be functioning as an activator as its loss leads to a reduction of expression of these genes. One stark exception is an operon coding for a bacterial microcompartment that metabolizes 1,2 propanediol for ATP production where LAerR appears to function as a repressor as a loss of LAerR leads to a rather dramatic increase in the expression of these bacterial microcompartment genes (*Supplementary file 4*). In the central metabolism category, many genes involved in glycolysis

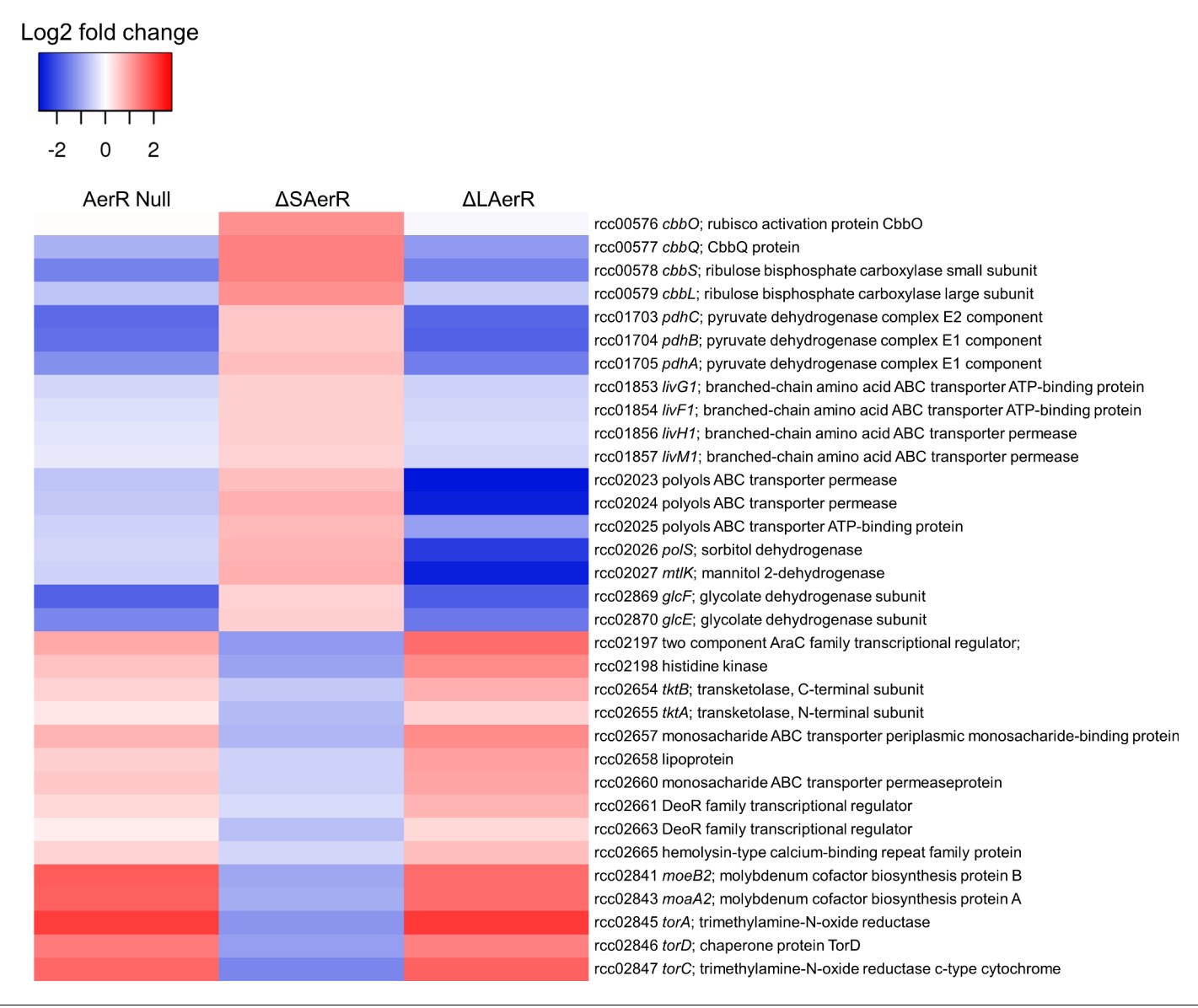

**Figure 7.** LAerR and SAerR regulates genes in opposite direction. Although most of the genes that are affected by disruption of LAerR do not change expression level upon disruption of SAerR, there are several genes that show an opposite change when SAerR is deleted. Examples include several carbon fixation genes, several branch chain transporters genes and TMAO reductase genes.

DOI: https://doi.org/10.7554/eLife.39028.014

The following figure supplement is available for figure 7:

**Figure supplement 1.** Regulation of CrtJ and AerR on the *liv3* operon.

DOI: https://doi.org/10.7554/eLife.39028.015

are reduced in both the AerR Null and LAerR depleted strains but increased in the SAerR depleted strain (*Supplementary file 4*). Specifically, these results indicate that LAerR activates expression of many glycolysis enzymes such as pyruvate dehydrogenase, fructose-bisphosphate aldolase (*fba*), glyceraldehyde-3-phosphate dehydrogenase (*gap1*), phosphoglycerate kinase (*pgk*) and pyruvate kinase (*pykA2*) (*Figures 7* and *8*). LAerR mediated increase in glycolysis likely lead to increased synthesis of pyruvate that feeds into the TCA cycle potentially increasing synthesis of a number of important molecules such as isoprenoids, tetrapyrroles and branched-chain amino acids (BCAA) (*Figure 8*). Conversely, increased expression of pyruvate dehydrogenase in the ΔSAerR strain indicates that the SAerR isoform has a role in decreasing the flow of metabolites into these cellular processes.

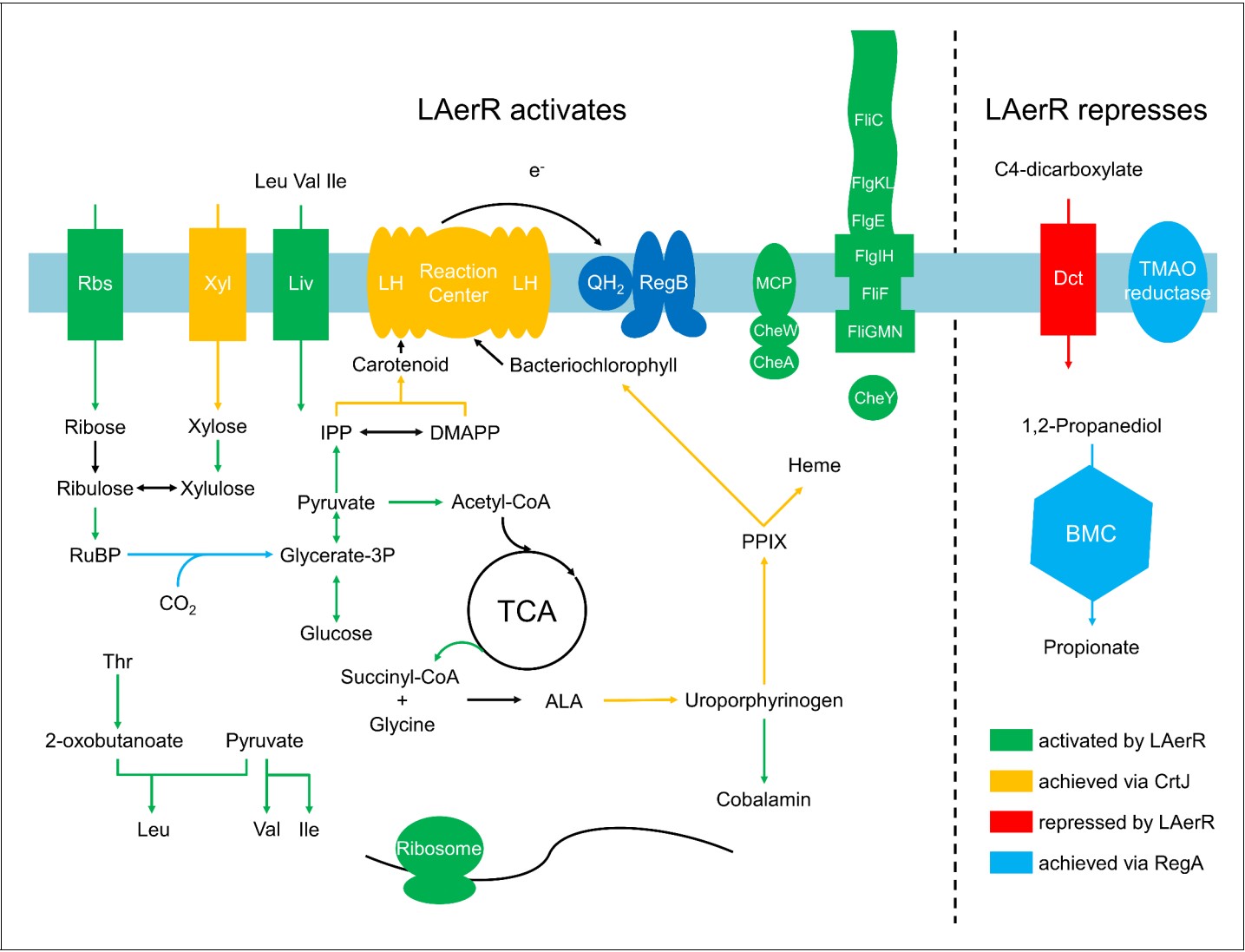

**Figure 8.** Summary of the LAerR regulon in *R. capsulatus* under photosynthetic growth conditions. LAerR activates carbon fixation, chemotaxis and motility, cobalamin biosynthesis, glycolysis and TCA cycle, heme biosynthesis, photosynthesis, ribosome and some transporters like xylose, ribulose and BCAA transporters. At the same time, LAerR represses bacterial microcompartment, TMAO reductase and C4-dicarboxylate transporter. These LAerR regulatory processes involve a direct interaction of LAerR with CrtJ.

DOI: https://doi.org/10.7554/eLife.39028.016

Interestingly, the same pattern of decreased expression by SAerR and increased expression by LAerR is observed for several genes involved in branched-chain amino acid transportation, carbon fixation mediated by form I (*cbbLS*) and form II RubisCO (*cbbM*) (**Figure 7**). Finally, LAerR stimulates expression of almost all the 50S and 30S ribosomal proteins, numerous genes involved in motility and chemotaxis, and numerous genes involved in cobalamin biosynthesis (**Figure 8** and **Supplementary file 4**). This latter effect is notable as LAerR itself uses cobalamin as a cofactor in a light-dependent manner.

## Discussion

In this study, we found that *R. capsulatus* synthesizes two isoforms of AerR, a full-length LAerR as well as a shorter SAerR variant that lacks 40 amino acids at the amino terminus. The ratio of the LAerR and SAerR isoforms changes as cells transition from logarithmic to stationary phases of growth with LAerR predominating in the log phase and SAerR predominating in the stationary

phase. The LAerR isoform also predominates in cells that are shifted from dark to illuminated photosynthetic conditions while the short form predominates when cells are shifted from light to dark conditions.

5′ race analysis shows two transcription start sites with a shorter transcript initiated within the LAerR coding region that is likely responsible for SAerR expression. Conversely, LAerR is expressed from a longer transcript, although at this stage we cannot rule out the possibility that the longer LAerR transcript is also responsible for both LAerR and SAerR expression. Our mutational studies also demonstrate that an alternative CTG (Leu41) translational start site is responsible for the production of SAerR. In bacteria, three initiation factors (IF1, IF2, and IF3) are involved in translation initiation of the ribosomal complex with IF3 inhibits translation from non-canonical initial codons such as ATT, ATC, and CTG. IF3 is known to inspect the initiation codon, thus in IF3 mutants there is increased translation from non-canonical codons (*Sussman et al., 1996*). In *R. sphaeroides*, a homolog of IF3 (PifC) was reported to possess similar activity as IF3 and also found to be an important factor for BchI biosynthesis and synthesis of the photosystem in semi-aerobic growth conditions (*Babic et al., 1997*). While it remains to be determined how changes in cellular growth rate/cellular physiology affects translation initiation at the CTG SAerR start site, it has been observed that many species sequesters 70S ribosomes into inactive 100S complexes in stationary phase with a protein known as YhbH (*Maki et al., 2000*; *Ueta et al., 2005*). Inactive 100S ribosomes can also be returned into an active 70S state with another factor termed YfiA when adequate growth conditions resume (*Maki et al., 2000*; *Ueta et al., 2005*). Orthologs of IF3, YhbH and YfiA are present in the *R. capsulatus* genome and thus would be good targets in future studies to explore the mechanism of how cellular physiology may ultimately control SAerR initiation at this internal CTG codon. We also observed a rapid reduction of LAerR in response to a growth shift from lit to dark conditions, which may indicate that LAerR is unstable under conditions where it would not contain bound cobalamin.

## AerR functions as an activation/repression switch of CrtJ function

An earlier in vitro study using small DNA templates for DNA binding analysis indicated that AerR (corresponding to LAerR in this study) likely functioned as an anti-repressor that dissociated CrtJ from photosystem promoters (*Cheng et al., 2014*). However, more recent in vitro studies, using much larger DNA segments, demonstrated that LAerR does not disassociate CrtJ from target promoters, but instead, alters CrtJ's interaction with the DNA template by significantly increasing the extent of the DNA that it interacts with (*Fang and Bauer, 2017*). Additional in vivo analysis using ChIP-seq also revealed that CrtJ does not significantly disassociate from target promoters under aerobic versus anaerobic conditions (*Fang and Bauer, 2017*). These results indicate that the control of photosystem gene expression by CrtJ is much more complex and nuanced than previously thought. Indeed in this study, we demonstrate that *R. capsulatus* synthesizes two isoforms of AerR, LAerR and SAerR, which have opposite effects on gene expression in a CrtJ dependent manner. As summarized in *Figure 9*, SAerR is the predominant variant in stationary phase under dark semi-aerobic conditions. Its interaction with CrtJ promotes CrtJ mediated super repression of the *bchEJGP, bchODI-crtA, crtIB* and *puc* operons leading to reduced synthesis of the photosystem (*Figure 9*). While CrtJ is capable of repressing photosystem gene expression on its own, its repression without SAerR seems partial or weaker than when CrtJ is complexed with SAerR. Previous studies have shown that CrtJ cooperatively binds to target promoters at tandem CrtJ binding motifs (TGT-N$_{12}$-ACA) (*Ponnampalam and Bauer, 1997*; *Elsen et al., 1998*; *Ponnampalam et al., 1998*). The CrtJ binding motifs are either located close together such as what occurs in the *bchC* promoter where they are 8 bp apart (*Ponnampalam et al., 1998*), or present at more distant locations 45 to 500 bp apart such as what occurs in the *puc, bchEJGP, bchODI-crtA, and crtIB* promoters (*Elsen et al., 1998*). In this regard, it's interesting that our RNA-seq results show that SAerR mediated enhancement of CrtJ repression is greater at promoters where CrtJ binding motifs are more distantly located than at the *bchC* promoter where CrtJ binding sites are only 8 bp apart.

In stark contrast to the ability of SAerR to enhanced CrtJ's ability to promote aerobic repression, our study also indicates that LAerR, which is the predominate isoform under photosynthetic conditions, switches CrtJ to an anaerobic photosynthetic activator (*Figure 9*). One clue to how this may occur is provided by previous in vitro and in vivo studies with LAerR which demonstrated that LAerR can dramatically affect CrtJ binding to the *bchC* promoter region (*Fang and Bauer, 2017*). Specifically, it was observed that CrtJ alone only bound to the two tandem *bchC* CrtJ binding motifs.

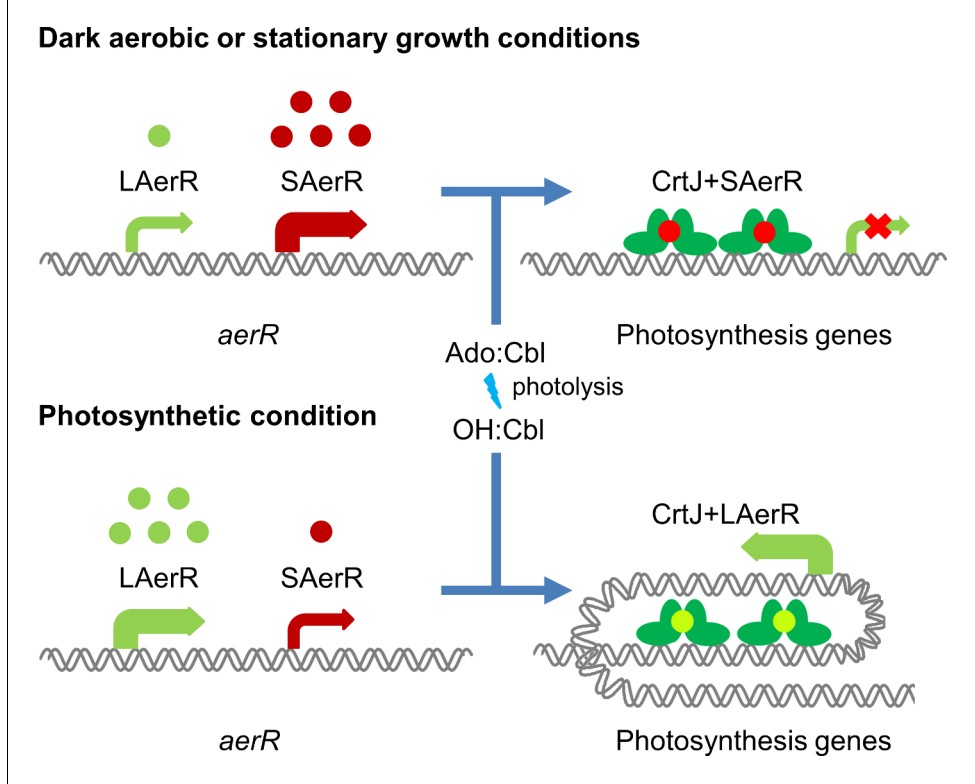

**Figure 9.** Model of LAerR and SAerR function in controlling gene expression. The results of this study, coupled with in vivo and in vitro DNA binding studies by Fang et al (*Fang and Bauer, 2017*), suggest that LAerR and SAerR are both capable of interacting with CrtJ. SAerR (red dot) complexed with Adenosyl- (Ado) or Methyl-cobalamin (Me:Cbl) is predominantly made under dark and/or stationary growth conditions. SAerR complexed with these biologically active cobalamin derivatives stimulates CrtJ (dark green) mediated photosystem repression. Under photosynthetic conditions, LAerR (light green dot) is the predominate form with this variant only binding hydroxyl cobalamin (OH-Cbl) that is generated via photolysis of Ado or methyl-cobalamin. LAerR bound with OH-Cbl interacts with CrtJ in a manner that promotes extensive interaction with target promoters (*Fang and Bauer, 2017*) activating gene expression.

DOI: https://doi.org/10.7554/eLife.39028.017

However when in the presence of LAerR, then CrtJ interacted with an extended region that spanned several hundred base pairs beyond the tandem CrtJ binding motifs (*Fang and Bauer, 2017*). We propose that LAerR mediated extension of CrtJ interaction to target promoters is likely responsible for switching CrtJ from an aerobic repressor to an anaerobic photosynthetic activator (*Figure 9*).

LAerR and SAerR also have clear differences in cobalamin binding characteristics with LAerR only binding OH-Cbl in a light-dependent manner while SAerR can bind all tested biologically relevant forms of cobalamin irrespective of light exposure. This result indicates that His10, which is only present in LAerR, is responsible for providing cobalamine specificity. LAerR also likely has a predominant role under illuminated conditions which is the growth condition that will form OH-Cbl as a product of photolysis of the upper ligand in Cbl. Conversely, the ability of SAerR to bind all tested forms of Cbl lends to its dominant role under dark conditions.

Differing roles of the LAerR and SAerR isoforms is also evident from RNA-seq results which show that LAerR controls expression of many more cellular process (photosystem synthesis and the expression of numerous glycolysis genes involved in central metabolism) than does SAerR. In WT cells the high LAerR/SAerR ratio during exponential phase presumably enhances energy production and protein biosynthesis needed for fast cellular growth. Conversely, the low LAerR/SAerR ratio observed in cells entering stationary phase would lead to reduced synthesis of the photosystem and reduced glycolysis as these cells are not actively replicating and thus require less energy production.

Finally, while LAerR likely has a direct effect on gene expression by interacting with CrtJ it also likely has an indirect role in controlling gene expression. For example, as diagramed in *Figure 8*,

deletion of LAerR leads to reduced synthesis of genes involved in energy production as well as reduced expression of many ribosomal genes. Indeed, reduced expression of numerous ribosomal genes in the ΔLAerR strain is puzzling as ribosomal genes show no in vivo binding of CrtJ (*Fang and Bauer, 2017*). One possible explanation is that the absence of LAerR in the ΔLAerR strain actually causes an energy limiting stringent like growth condition which is known to lead to reduced expression of ribosomal genes. Along this vein, the branch chain amino acid (BCAA) transporter genes (*rcc03426-03433*), that code for an importer of BCAA's, were found to be highly directly downregulated in the AerR null and in the ΔLAerR strains (*Figure 7—figure supplement 1*). Furthermore, several BCAA were recently shown to stimulate the degradation of the cellular alarmone molecules (p) ppGpp which regulate the stringent response that is known to influence expression of ribosomal genes (*Fang and Bauer, 2018*). Another consequence of reduced synthesis of the photosystem by the ΔLAerR strain would be fewer electrons flowing from the photosystem to the quinone pool. Such an alteration in the redox state of ubiquinones would be sensed by the ubiquinone responding sensor kinase RegB leading to downstream alterations in global gene expression by RegA (*Wu and Bauer, 2010*; *Schindel and Bauer, 2016*). This might be the reason why part of RegA regulon (chemotaxis and motility genes, bacterial microcompartment etc.) is also observed to be indirectly affected upon deletion of AerR.

## L and SAerR isoforms are also present in other photosynthetic bacteria

It is known that the AerR gene is present and linked to the CrtJ gene in all sequenced purple non-sulfur bacteria (*Cheng et al., 2014*; *Vermeulen and Bauer, 2015*). The long isoform of AerR has also been isolated from numerous species and shown to bind cobalamin (*Vermeulen and Bauer, 2015*). In a previous study, AerR and CrtJ from *Rhodospirillum centenum* were also disrupted with resulting phenotypes indicating that AerR and/or CrtJ likely have dual functions for both activation and repression of photosystem gene expression (*Masuda et al., 2008*). Our results are in good agreement with the *R. centenum* study. Furthermore, we have also observed that *R. centenum* also synthesize both long and short isoforms of AerR (*Figure 1—figure supplement 2*).

While the presence of two isoforms in other species suggests that diverse species of purple bacteria may use similar LAerR/SAerR isoforms to control gene expression, it does not indicate that they all do so. For example, the AerR homolog from *Rhodobacter sphaeroides* (also called PpaA) appears to only have one large isoform (*Figure 1—figure supplement 2*). Interestingly, this species synthesizes a second photoreceptor protein called AppA that contains extensive homology to AerR with the caveat that AppA uses flavin as a chromophore instead of cobalamin (*Moskvin et al., 2010*). A study has also shown that AerR and AppA have opposing functions in regulating the activity of CrtJ (*Vermeulen and Bauer, 2015*) suggesting that this species may have replaced the short AerR isoform with a gene duplication event that gave rise to AppA.

## Materials and methods

**Key resources table**

| Reagent type (species) or resource | Designation | Source or reference | Identifiers | Additional information |
|---|---|---|---|---|
| Genetic reagent (*Rhodobacter capsulatus*) | WT | PMID: 1262313 | Wild type parent strain | This laboratory |
| Genetic reagent (*Rhodobacter capsulatus*) | ΔLAerR | This study | | Strain that only expresses SAerR |
| Genetic reagent (*Rhodobacter capsulatus*) | ΔSAerR | This study | | Strain that only expresses LAerR |
| Genetic reagent (*Rhodobacter capsulatus*) | AerR null | This study | | Strain that does not express either L/SAerR |
| Genetic reagent (*Rhodobacter capsulatus*) | AerR 3xFLAG | PMID: 28325764 | | Strain with *aerR* 3xFLAG on 3' |
| Genetic reagent (*Rhodobacter capsulatus*) | ΔLAerR 3xFLAG | This study | | Strain that only expresses SAerR in AerR 3xFLAG background |

*Continued on next page*

*Continued*

| Reagent type (species) or resource | Designation | Source or reference | Identifiers | Additional information |
|---|---|---|---|---|
| Genetic reagent (*Rhodobacter capsulatus*) | ΔSAerR 3xFLAG | This study | | Strain that only expresses LAerR in AerR 3xFLAG background |
| Genetic reagent (*Rhodobacter capsulatus*) | AerR null 3xFLAG | This study | | Strain that does not express either L/SAerR in AerR 3xFLAG background |
| Genetic reagent (*Rhodobacter capsulatus*) | ΔcrtJ | PMID: 28325764 | | Strain that does not express CrtJ |
| Genetic reagent (*Rhodobacter capsulatus*) | ΔLAerR ΔcrtJ | This study | | Strain that only expresses SAerR in ΔcrtJ background. |
| Genetic reagent (*Rhodobacter capsulatus*) | ΔSAerR ΔcrtJ | This study | | Strain that only expresses LAerR in ΔcrtJ background. |
| Genetic reagent (*Rhodobacter capsulatus*) | AerR null ΔcrtJ | This study | | Strain that does not express either L/SAerR in ΔcrtJ background. |
| Genetic reagent (*Escherichia coli*) | BL21 (DE3) | NEB | C2527 | |
| Genetic reagent (*Escherichia coli*) | S17-1λpir | PMID: 6340113 | | |
| Genetic reagent (*Escherichia coli*) | HST08 | TaKaRa Bio | 9128 | |
| antibody | Anti-FLAG epitope monoclonal antibody HRP conjugate | Sigma | A8592 | |
| recombinant DNA reagent | pSUMO-CrtJ | PMID: 22715852 | | |
| recombinant DNA reagent | pSUMO-SAerR | This study | | Plasmid that express SUMO-SAerR under T7 promotor. |
| recombinant DNA reagent | pSUMO-AerR | PMID: 24329562 | | Plasmid that express SUMO-LAerR under T7 promotor. |
| recombinant DNA reagent | pBBR-aerR 3xFLAG | This study | | Plasmid that express *aerR*-3xFLAG |
| recombinant DNA reagent | pBBR-aerR_Met1a | This study | | Plasmid that express *aerR*-3xFLAG with a nucleotied A insertion after the Met1 codon |
| recombinant DNA reagent | pBBR-aerR_Met35 | This study | | Plamid that express *aerR*-3xFLAG with a truncation from the Met1 codon to the 34th codon |
| recombinant DNA reagent | pBBR-aerR_Met49 | This study | | Plamid that express *aerR*-3xFLAG with a truncation from the Met1 codon to the 48th codon |
| recombinant DNA reagent | pBBR-aerR_Met61 | This study | | Plamid that express *aerR*-3xFLAG with a truncation from the Met1 codon to the 60th codon |

*Continued on next page*

Continued

| Reagent type (species) or resource | Designation | Source or reference | Identifiers | Additional information |
|---|---|---|---|---|
| recombinant DNA reagent | pBBR-aerR_M35 + 1 | This study | | Plasmid that express *aerR*-3xFLAG with a nucleotied A insertion after the Met35 codon |
| recombinant DNA reagent | pBBR-aerR_V38 + 1 | This study | | Plasmid that express *aerR*-3xFLAG with a nucleotied A insertion after the Val38 codon |
| recombinant DNA reagent | pBBR-aerR_L41 + 1 | This study | | Plasmid that express *aerR*-3xFLAG with a nucleotied A insertion after the Leu41 codon |
| recombinant DNA reagent | pBBR-aerR_T43 + 1 | This study | | Plasmid that express *aerR*-3xFLAG with a nucleotied A insertion after the Thr43 codon |
| recombinant DNA reagent | pBBR-aerR_V44 + 1 | This study | | Plasmid that express *aerR*-3xFLAG with a nucleotied A insertion after the Val44 codon |
| recombinant DNA reagent | pBBR-aerR_A39 -FLAG-bchE | This study | | Plasmid that express 3xFLAG-*bchE* with a partial *aerR* sequence from M1 to A39 as a promotor and an initiation codon |
| recombinant DNA reagent | pBBR-aerR_E40 -FLAG-bchE | This study | | Plasmid that express 3xFLAG-*bchE* with a partial *aerR* sequence from M1 to E40 as a promotor and an initiation codon |
| recombinant DNA reagent | pBBR-aerR_L41 -FLAG-bchE | This study | | Plasmid that express 3xFLAG-*bchE* with a partial *aerR* sequence from M1 to L41 as a promotor and an initiation codon |
| peptide, recombinant protein | SUMO-CrtJ | PMID: 22715852 | | |
| peptide, recombinant protein | SUMO-LAerR | PMID: 24329562 | | |
| peptide, recombinant protein | SUMO-SAerR | This study | | SUMO-tagged SAerR |
| commercial assay or kit | In Fusion HD cloning kit | Clontech | 639648 | |
| commercial assay or kit | GeneRacer Kit | Invitrogen | 150201 | |
| commercial assay or kit | Terminator exonuclease | Epicentre | TER51020 | |
| commercial assay or kit | MST labeling kit | NanoTemper | MO-L001 | |
| software, algorithm | Trimmomatic | PMID: 24695404 | | |
| software, algorithm | Bowtie2 | PMID: 22388286 | | |
| software, algorithm | HTSeq | PMID: 25260700 | | |
| software, algorithm | DESeq2 | PMID: 20979621 | | |

## Strains and cultivation

The *Rhodobacter capsulatus* strain SB1003 was used as the WT parental strain and was also the host strain from which L*aerR* and S*aerR* expression strains were constructed. *R. capsulatus* strains were first grown semi-aerobically overnight as a 3 ml PY medium in tubes at 34°C with shaking at 200 rpm. The overnight cultures were then transferred to flasks shaking at 200 rpm for aerobic conditions or into screw-caped vials for anaerobic conditions. 75 W tungsten filament light bulbs were used as a light source under anaerobic photosynthetic conditions. *E. coli* strains, HST08 and S17-1λpir were used for cloning and for the conjugation of plasmids to *R. capsulatus*, respectively. AerR overexpression was carried out using *E. coli* strain BL21 (DE3) grown in LB medium. UNICO 1100RS Spectrometer was used to check growth curves under photosynthetic conditions.

## Plasmid and strains construction

To express AerR-FLAG protein in *R. capsulatus* cells, we constructed an in-frame chromosomal FLAG-tagged *aerR* strain (*Fang and Bauer, 2017*) as well as expressed AerR from a low copy broad-host range vector, pBBR-MSC2. For the plasmid construction, DNA fragments containing the *aerR* coding region with an appended FLAG-tag sequence (Rc_aerR-f and pSRK-pBBR-r) were amplified along with 500 bp upstream and downstream of the *aerR* gene (Rc_aerRup-f and Rc_aerRup-r) using the primers in *Supplementary file 5* from pSRKGm-aerR (*Fang and Bauer, 2017*) and genomic DNA from *R. capsulatus as a template*. These two DNA fragments were connected and cloned into pBBR-MSC2 *Eco*RV and *Hind*III site using In Fusion cloning kit (Clonetech). To construct an N-terminal truncated AerR expression plasmid, Rc_aerR-insA-f (one nucleotide insertion after M1 codon), Rc_aerR_MNG-f (M35-AerR), Rc_aerR_MVE-f (M49-AerR), or Rc_aerR_MDL-f (M61-AerR) primers were used instead of Rc_aerR-f. Each point mutation was introduced into the pBBR-aerR-FLAG plasmid by PCR amplification using specific primer pairs (*Supplementary file 5*). For a reporter assay, FLAG-bchE fragment was amplified from pSRKGm-bchE plasmid with Flag-bchE-f and M13+ pBBR-MSC2-r and pBBR plasmid that included a partial aerR sequence. This was amplified with pBBR-MSC2-f and Rc_aerRA39/E40/L41-FbchE-r, respectively from the pBBR-aerR-FLAG plasmid. Then, these two fragments were connected using an In-Fusion kit (Clonetech), resulting pBBR-aerR_A39-FLAG-bchE, pBBR-aerR_E40-FLAG-bchE, and pBBR-aerR_L41-FLAG-bchE, respectively. Chromosomal *aerR* mutations were generated using the suicide plasmid pZJD29a containing 1 kb fragment covering *aerR* gene with the point mutation as previously reported (*Cheng et al., 2014*). The *aerR* fragment with select mutations were amplified from the corresponding pBBR-aerR-FLAG plasmid using the primer pairs Rc_aerRup-f2 and Rc_aerR1130-r) and cloned into pZJD29a using the In Fusion kit. For in vitro analysis, pSUMO-AerR (*Cheng et al., 2014*) was used to express LAerR isoform in *E. coli*. A 120 bp DNA sequence including M1 to E40 codons of *aerR* was removed from pSUMO-AerR plasmid to express the SAerR isoform. The deletion was made by PCR amplification using Rc_saerR + SUMO f and Rc_saerR + SUMO r primers using pSUMO-AerR as a template with the resulting plasmid, pSUMO-SAerR transformed BL21 (DE3) to express the SAerR isoform.

## Western blot analysis

In vivo expression of AerR proteins in *R. capsulatus* cells were measured by Western blot analysis after the addition FLAG epitope to the carboxyl terminus of AerR. For this analysis, collected *R. capsulatus* cells were resuspended in TBS buffer and then disrupted by sonication. Disrupted cell extracts were clarified by centrifugation 20,000 x g for 10 min at 4°C. Clarified proteins in the supernatant were separated by SDS-PAGE followed by Western blot analysis that detected the FLAG epitope using commercial FLAG epitope-specific monoclonal antibodies containing an HRP conjugate (Sigma).

## RNA extraction and RNA-seq

*R. capsulatus* strains were grown to early exponential phase ($OD_{660}$0.3–0.35) from which 1.5 ml of cell cultures were quickly chilled to 4°C, harvested by centrifugation and stored as a cell pellet at −80°C until needed. Triplicate biological replicates (independent cell cultures grown under similar conditions at different times) were used for each RNA-Seq analysis for each described condition. Total RNA was extracted using ISOLATE II RNA Mini Kit (Bioline) followed by TURBO DNase (Ambion) treatment. The reaction mixture was cleaned and concentrated by RNeasy MinElute

Cleanup Kit (QIAGEN) and assayed for DNA contamination by PCR amplification on samples with or without reverse transcriptase treatment. Final RNA concentrations were measured using NanoDrop spectroscopy (Thermo Scientific). Further quality control was performed with a 2200 TapeStation using RNA ScreenTape (Agilent Technologies). Library construction and RNA-sequencing were performed by the Center for Genomics and Bioinformatics at Indiana University-Bloomington. Ribosomal RNA was depleted and libraries were created using a ScriptSeq Complete Kit (Illumina) for bacteria according to manufacturer's protocol. Single-end sequencing reactions (>75 $\times$ coverage) were performed on Illumina NextSeq sequencer with raw sequence read files deposited in Sequence Read Archive (SRA) with the accession number SRP136743. The raw reads were trimmed and aligned to the *R. capsulatus* SB1003 annotated genome (GenBank accession no. CP001312.1) as described previously using Bowtie 2 (*Langmead and Salzberg, 2012*). HTSeq-count (*Anders et al., 2015*) was used to count read numbers in each gene followed by differential expression analysis using DESeq2 package in R (*Love et al., 2014*). Genes were considered to be significantly different if they had a p-adjusted value <0.01.

## 5'-RACE

Total RNA that was extracted from the previous RNA-seq step was used for RNA ligase-mediated rapid amplification of 5' cDNA ends (RLM-RACE). GeneRacer Kit (Invitrogen) was used to generate RACE-ready cDNA, except that the calf intestinal alkaline phosphatase (CIP) treatment was replaced by Terminator exonuclease (Epicentre) treatment in order to select for primary transcripts. GeneRacer 5' primer and an *aerR* specific primer were used in 5' RACE PCR. RACE PCR product was gel purified and cloned into pCR-4 TOPO (Invitrogen). 20 clones were selected for sequencing to validate the transcription start site of *aerR*.

## Protein purification

For biochemical analysis, AerR variant proteins were purified as described previously (*Cheng et al., 2014*). *E. coli* strain BL21 (DE3) with pSUMO-AerR or pSUMO-SAerR was grown in LB medium at 37°C to an OD600 of 0.7. *E. coli* cultures were then cooled and then 50 µM isopropyl-β-D-thiogalactopyranoside (IPTG) was added with cultivation continued at 16°C for 16 hr. Collected cells was resuspended in a lysis buffer (20 mM Tris-HCl pH 8.0, 150 mM NaCl, 5 mM imidazole, and 10% glycerol) and disrupted using a French press cell three times at 18,000 psi. The lysate was clarified by centrifugation at 30,000 x g for 30 min at 4°C. To bind hydroxyl-Cbl, 10 µM adenosyl-Cbl was added to the supernatant followed by illumination with white light for 5 min. The supernatant was then passed through a 0.45 µM membrane filter and applied to a 1 ml HisTrap column using ÄKTA chromatography system. The column was washed with wash buffer (20 mM Tris-HCl pH 8.0, 150 mM NaCl, 20 mM imidazole, and 10% glycerol) and SUMO-AerR was eluted with a gradient of 20 mM to 500 mM imidazole in the wash buffer over 15 column volume. Eluted SUMO-AerR was incubated with SUMO protease Ulp1 in presence of 1 mM DTT at RT for 16 hr followed by a desalting column against the wash buffer. Digested SUMO-tag was trapped by Ni-sepharose column with tag-less AerR further purified by Superose 12 size exclusion chromatography in 20 mM Tris-HCl (pH 8.0) and 200 mM NaCl.

Microscale thermophoresis (MST) analysis was performed using Monolith NT.115 (Nanotemper) as described previously (*Cheng et al., 2014*). CrtJ was labeled using RED-NHS protein labeling kit (Nanotemper) and the labeling efficiency was evaluated spectroscopically. For the MST experiment, concentration of labeled CrtJ was kept constant (either 200 or 500 nM) in 20 mM Tris-HCl (pH 8.0) and 200 mM NaCl and with AerR concentration varied from µM to nM scale.

Cobalamin binding assay was performed using cobalamin unbound SUMO-tagged AerR protein. Cobalamin unbound SUMO-(S)AerR was purified as same as mentioned above with the exception of cobalamin addition to cell lysates followed by incubation for 20 min at RT under dark conditions. Unbound cobalamin molecules were removed by desalting column. Cobalamin binding was evaluated by the spectrum of the SUMO-(S)AerR fraction.

## Pigment analysis

Total pigment was extracted from *R. capsulatus* cell by acetone/methanol (v:v = 7:2). Collected cells were dissolved in the acetone/methanol solution, followed by cell disruption using sonication. After

the extract was clarified by centrifugation (13,300 rpm, 10 min, 4°C), absorption spectrum was scanned from 350 nm to 900 nm with a HP 8453 UV-Visible Spectrometer. Relative amounts of bacgteriochlorophyll and carotenoid were calculated from absorbance at 768 nm, and 480 nm, respectively.

## Acknowledgements

We thank the staff at the Indiana University Center or Genomics and Bioinformatics for their help in library construction and deep sequencing. This work was supported by a National Institutes of Health grant GM040941 awarded to CEB.

## Additional information

### Funding

| Funder | Grant reference number | Author |
| --- | --- | --- |
| National Institutes of Health | GM040941 | Carl E Bauer |

The funders had no role in study design, data collection and interpretation, or the decision to submit the work for publication.

### Author contributions

Haruki Yamamoto, Conceptualization, Formal analysis, Validation, Investigation, Methodology, Writing—original draft; Mingxu Fang, Conceptualization, Data curation, Formal analysis, Validation, Investigation, Methodology, Writing—original draft; Vladimira Dragnea, Investigation, Methodology, Writing—review and editing; Carl E Bauer, Conceptualization, Formal analysis, Supervision, Funding acquisition, Investigation, Project administration, Writing—review and editing

### Author ORCIDs

Mingxu Fang (iD) https://orcid.org/0000-0003-1595-8046
Carl E Bauer (iD) http://orcid.org/0000-0002-1432-0756

### Decision letter and Author response

Decision letter https://doi.org/10.7554/eLife.39028.028
Author response https://doi.org/10.7554/eLife.39028.029

## Additional files

### Supplementary files

• Supplementary file 1. This file shows expression for all genes in the *R. capsulatus* genome listed in numerical order for the AerR null strain, the strain lacking SAerR and the strain lacking LAerR. The expression changes were determined relative to expression levels observed in the parent wild type strain using RNA-seq. Cells were grown under dark semi-aerobic conditions. p-adjusted value <0.01 are shown in red.
DOI: https://doi.org/10.7554/eLife.39028.018

• Supplementary file 2. This file shows expression for all genes in the *R. capsulatus* genome listed in numerical order for the AerR null strain, the strain lacking SAerR and the strain lacking LAerR. The expression changes were determined relative to expression levels observed in the parent wild type strain using RNA-seq. Cells were grown under illuminated anaerobic photosynthetic conditions. p-adjusted value <0.01 are shown in red.
DOI: https://doi.org/10.7554/eLife.39028.019

• Supplementary file 3. This file contains a list of all genes in the AerR null strain that undergo significant changes in expression as based on a p-adjusted value <0.01. The gene expression changes are shown as log 2-fold changes and were measured using RNA-seq. The first tab shows all genes

undergoing changes in expression in cells grown dark semi-aerobically while the second tab shows changes in expression in cells grown under illuminated anaerobic photosynthetic conditions.
DOI: https://doi.org/10.7554/eLife.39028.020

• Supplementary file 4. This file contains a list of all genes that undergo significant changes in expression as based on a p-adjusted value <0.01. The gene expression changes are shown as log 2-fold changes and were measured using RNA-seq. The strains assayed are the AerR null strain, a strain lacking SAerR (DSAerR) and a strain lacking LAerR (DLAerR). All strains were grown under illuminated anaerobic photosynthetic conditions. Note that genes are clustered into groups of similar function with individual tabs.
DOI: https://doi.org/10.7554/eLife.39028.021

• Supplementary file 5. This file lists the sequence of all PCR primers used in this study for plasmid and mutant construction.
DOI: https://doi.org/10.7554/eLife.39028.022

• Transparent reporting form
DOI: https://doi.org/10.7554/eLife.39028.023

## Data availability

RNA-seq sequence read files have been deposited in Sequence Read Archive (SRA) with the accession number SRP136743 and can be accessed at the URL: https://trace.ncbi.nlm.nih.gov/Traces/sra/?study=SRP136743.

The following dataset was generated:

| Author(s) | Year | Dataset title | Dataset URL | Database, license, and accessibility information |
|---|---|---|---|---|
| Fang M, Bauer CE | 2017 | Functional analyses of long and short AerR in Rhodobacter capsulatus | https://trace.ncbi.nlm.nih.gov/Traces/sra/?study=SRP136743 | Publicly available at NCBI BioProject (accession no: SRP136743) |

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
