## [Decision Letter]

Thank you for submitting your article "Differing Isoforms of the Cobalamin Binding Photoreceptor AerR Oppositely Regulate Photosystem Expression" for consideration by *eLife*. Your article has been reviewed by three peer reviewers, and the evaluation has been overseen by Gisela Storz serving both as Reviewing and Senior Editor. The following individual involved in review of your submission has agreed to reveal his identity: Alexander Mankin (Reviewer #3).

The reviewers have discussed the reviews with one another and the Reviewing Editor has drafted this decision to help you prepare a revised submission. The list of revisions is on the long end for *eLife* but the hope is that most will be quick to address.

Summary:

The paper of Yamamoto et al. presents a finding that the aerR gene in the purple photosynthetic bacterium *Rhodobacter capsulatus* is translated from two start codons leading to the production of longer and shorter isoforms of the AerR protein (LAerR and SAerR, respectively). The isoforms are differentially expressed in cells grown under different conditions and differ in their ability to interact with cobalamine. They also differ in their effect on gene regulation, likely mediated by the transcriptional repressor CrtJ.

Essential revisions:

1) The demonstration that the aerR gene is expressed from two start codons is the key finding of the study. These data need to be in the main text.

2) It is unclear whether switch in expression of the long vs. short forms of AerR is controlled at the level of transcription or translation. This would largely depend on whether both protein isoforms could be expressed from the long mRNA. The authors should address this by expressing long mRNA (prepared by in vitro transcription from a T7 promoter) on the bacterial (e.g. *E. coli*) cell-free system (PUREsystem, NEB or cell extract based systems from Promega or Roche) and test whether one or two protein isoforms are generated. The reviewers understand that if the experiment shows that both proteins are made from the long in vitro transcript, this is a valuable information. If not, this may well be due to differences in translation between *E. coli* and *Rhodobacter* or the in vitro and in vivo situation.

3) The results of the RNAseq experiment need to be confirmed for some selected genes by another method. In case of the ΔAerR strain under dark conditions the changes are limited and restricted to certain functional groups. However, under phototrophic conditions change for > 1500 genes is observed. This may be a real physiological response but it is important to exclude any artifacts, which may e.g. stem from calibration problems.

4) AerR-CrtJ binding may be influenced by different cobalamin derivatives. Thus, the experiment in Figure 5 should be done with different cobalamin derivatives; this should provide important information of how SAerR and LAerR differentially control CrtJ activity.

5) The authors should perform carotenoid determination in photosynthetically grown cells. Furthermore, such pigment analysis may need to be performed at different growth phases to test how LAerR levels influences pigmentation. The authors should also characterize the absorption spectra of membrane preparations of the mutants. Such data will provide useful information of how SAerR and LAerR differentially regulate puf and puc operons.

6) The authors should point out the spectral features of Abo-Cbl and OH-Cbl bound SAerR were actually different. This spectral change reflects difference of chromophore-protein interactions between Abo-Cbl and OH-Cbl bound SAerRs; thus the two protein isoforms are in different conformations. Biochemical analysis of AerR (e.g. as in Figure 5) needs to be done with, at least, the two isoforms.

7) Some results need further explanation/details:

- The authors indicate that 5'RACE showed two transcription initiation sites (Discussion, second paragraph and supplementary translation data), but the gel in Figure 2—figure supplement 1 shows only one PCR band. It is unclear which PCR product was sequenced to identify the second transcription start site.

- The authors suggest that differential translation is driven by differential transcription. Does their RNA-seq data show an increase in the aerR gene coverage after the second transcription start site.

- Figure 3 and 6: The details of the cell culture conditions were not provided: e.g. the cells were harvested at mid-log phase? Because the level of LAerR isoform is different in each growth phase (Figure 1), the information should be given.

- Figure 4: The observed growth delay for dLAerR and AerR-null mutants seems to result from low-pigmentation under aerobic conditions before starting the measurement (Figure 3A).

- The details of the experimental conditions for the AerR and CrtJ interaction (Figure 5) were not provided. For example, it is not clear whether the AerR used in the experiment was cobalamin-bound form or unbound form. Please provide such information. If the authors used cobalamin-bound form, what kind of cobalamin bound to the apo-AerR?

- The SAerR variant, which does no longer bind cobalamin due to point mutations, still has a strong phenotype (spectra and growth). It would be appropriate to comment on this instead of claiming only slight effects.

- It is possible that the AerR point mutations influences not only cobalamin-binding characteristics, but also stability/level of AerR in vivo. The authors should comment on this.

- LAerR, but not SAerR, expression level is changed in response to growth modes (Figure 1), and SAerR can bind both photo-activated and non-activated cobalamin although LAerR can bind only photo-activated one (Figure 7). Although the RNA seq gave some information about the functional differences of the two isoforms (Figure 9, 10), the biochemical properties of the two AerR isoforms is not clearly discussed in the manuscript. The manuscript would benefit from such discussion, with schematic models of the action of each AerR isoform and CrtJ.

8) Some text needs to be modified:

- It is sometimes hard to follow the story and to extract the important messages. The authors should make an effort for the text to be accessible to a wide audience.

- It is not clear from the Abstract why photosynthetic bacteria control photosystem synthesis. What is the meaning of the first sentence of the Abstract? What do the authors mean by "adjusts their physiology for optimal light absorption"?

In the first paragraph of the Introduction, the authors briefly mention that photosystems can generate oxidative stress, and that photosystem production must also be controlled to avoid oxidative stress. The important function of photoreceptors to monitor conditions, which may cause photooxidative stress and to trigger the crucial adaptation process to these conditions is not really considered in other parts of the paper.

- The discussion of the IF3 function (Discussion, second paragraph) is inaccurate. IF3 is not 'involved in translation from non-canonical initiation codons' and it does not 'increase initiation from non-canonical codons'. Instead, IF3 discriminates against the non-canonical initiation codons.

- The discussion of the hibernation factors and formation of the 100S ribosomes in the stationary phase seem irrelevant for their findings.

---

## [Author Response]

Essential revisions:1) The demonstration that the aerR gene is expressed from two start codons is the key finding of the study. These data need to be in the main text.

Even though moving this data from Supplementary files to the Results section increases the page length by three pages, we do agree that its appropriate to do so. Consequently, this data has been moved into the Results section.

2) It is unclear whether switch in expression of the long vs. short forms of AerR is controlled at the level of transcription or translation. This would largely depend on whether both protein isoforms could be expressed from the long mRNA. The authors should address this by expressing long mRNA (prepared by in vitro transcription from a T7 promoter) on the bacterial (e.g. E. coli) cell-free system (PUREsystem, NEB or cell extract based systems from Promega or Roche) and test whether one or two protein isoforms are generated. The reviewers understand that if the experiment shows that both proteins are made from the long in vitro transcript, this is a valuable information. If not, this may well be due to differences in translation between E. coli and Rhodobacter or the in vitro and in vivo situation.

Actually, we have expressed the AerR-FLAG tagged gene in *E. coli* in vivo using a T7 expression system. Western blot analysis shows that only the full-length long isoform is produced. This result would be no different than what would be observed in a cell free system. Note also that *R. capsulatus* has a very high GC content so its native -10 promoter recognition sequences are quite different than that of typical *E. coli* promoters. Consequently, its highly likely that *E. coli* polymerase does not recognize these two R. capsulatus AerR promoters.

That said, we do provide evidence that both transcription and translation contributed to the control of LAerR vs. SAerR. For example, we have observed that we can significantly increase SAerR protein levels by just mutating its start codon CTG to a silent Leu TTG (Figure 3—figure supplement 1), this is an evidence that translation initiation indeed plays an role in controlling SAerR levels. We also showed that a truncation (Figure 2—figure supplement 1) construct without the promoter for the short transcript still produced both LAerR and SAerR, indicating that a single long transcript is capable of synthesizing both isoforms (Figure 2—figure supplement 1A).

3) The results of the RNAseq experiment need to be confirmed for some selected genes by another method. In case of the ΔAerR strain under dark conditions the changes are limited and restricted to certain functional groups. However, under phototrophic conditions change for > 1500 genes is observed. This may be a real physiological response but it is important to exclude any artifacts, which may e.g. stem from calibration problems.

We do appreciate the question about validation of our RNA seq data. Note that our laboratory has a lengthy publication record with numerous studies that analyzed global expression changes via RNA-seq (1-6). In all of our RNA-seq studies we validate our results using qPCR with a variety of select target genes. We also note that observed changes in carotenoid and bacteriochlorophyll levels also faithfully mirrors changes that we see in RNA transcript reads that we obtain via RNA-seq. Consequently, we are highly confident of the transcription values obtained via our in house RNA-seq analysis pipeline. In addition, previously published ChIP-seq data showing colocalization of CrtJ and AerR to operons that are highly regulated by these regulators also correlates well with the RNA-seq data reported in this study. This latter confirmation is shown in Figure 7—figure supplement 1.

1) Dong, Q., and C. E. Bauer (2015) Transcriptome analysis of cyst formation in Rhodospirillum centenum reveals large global changes in expression during cyst development. BMC Genomics. 16:68 DOI: 10.1186/s12864-015-1250-9. PMCID: PMC4340629

2) Dong, Q., Roychowdhury, S., and C. E. Bauer (2015) Mapping The CgrA Regulon Of Rhodospirillum centenum Reveals a Hierarchal Network Controlling Cyst Development. BMC Genomics. 16:1066. PMCID: PMC4681086

3) Kumka J. E. and C. E. Bauer(2015) Analysis of the FnrL regulon in Rhodobacter capsulatus reveals limited regulon overlap with orthologues from Rhodobacter sphaeroides and Escherichia coli. BMC Genomics 16:895 doi:10.1186/s12864-015-2162-4. PMID: 26537891

4) Kumka JE, Schindel H, Fang M, Zappa S, and Bauer CE (2017) Transcriptomic analysis of aerobic respiratory and anaerobic photosynthetic states in Rhodobacter capsulatus and their modulation by global redox regulators RegA, FnrL and CrtJ. Microbial genomics 3(9):e000125.

5) Schindel HS and Bauer CE (2016) The RegA regulon exhibits variability in response to altered growth conditions and differs markedly between Rhodobacter species. Microbial genomics 2(10):e000081.

6) Malinich, E. and C. E. Bauer (2018) Transcriptome Analysis Of Azospirillum brasilense Vegetative And Cyst States Reveals Large Scale Alterations In Metabolic And Replicative Gene Expression. Microbial Genomics (In Press)

4) AerR-CrtJ binding may be influenced by different cobalamin derivatives. Thus, the experiment in Figure 5 should be done with different cobalamin derivatives; this should provide important information of how SAerR and LAerR differentially control CrtJ activity.

As requested, we re-did the thermophoresis binding analysis with OH-Cbl bound SAerR as well as with Ade-Cbl bound SAerR. The binding results were virtually the same with both of these variants as well with that observed with LAerR. These new binding results are now presented in the Results section.

5) The authors should perform carotenoid determination in photosynthetically grown cells. Furthermore, such pigment analysis may need to be performed at different growth phases to test how LAerR levels influences pigmentation. The authors should also characterize the absorption spectra of membrane preparations of the mutants. Such data will provide useful information of how SAerR and LAerR differentially regulate puf and puc operons.

Carotenoid determination is now provided in a revised Figure 3. Differential effects of SAerR and LAerR on expression of the puf and puc operons is also found in the RNA-seq data.

6) The authors should point out the spectral features of Abo-Cbl and OH-Cbl bound SAerR were actually different. This spectral change reflects difference of chromophore-protein interactions between Abo-Cbl and OH-Cbl bound SAerRs; thus the two protein isoforms are in different conformations. Biochemical analysis of AerR (e.g. as in Figure 5) needs to be done with, at least, the two isoforms.

The spectra of Ado-Cbl and OH-Cbl bound SAerR were indeed different, however, this is actually do to spectral differences of Ado-Cbl and OH-Cbl. Indeed, all of the various Cbl derivatives have different spectral characteristics owing to these different upper axial ligands. To make it clear that SAerR indeed binds each of these Cbl we have added an additional supplemental spectrum (Figure 1—figure supplement 2).

7) Some results need further explanation/details:- The authors indicate that 5'RACE showed two transcription initiation sites (Discussion, second paragraph and supplementary translation data), but the gel in Figure 2—figure supplement 1 shows only one PCR band. It is unclear which PCR product was sequenced to identify the second transcription start site.

The gel matrix that showed a 5’RACE product was not designed to resolve two similar sized bands. It was only shown to verify that we have a 5’RACE product(s). In hindsight, we should not have added this gel to Figure 2—figure supplement 1 as it was confusing to the reader. Note that sequence analysis of cloned the 5’RACE products clearly showed the presence of two transcription start sites.

- The authors suggest that differential translation is driven by differential transcription. Does their RNA-seq data show an increase in the aerR gene coverage after the second transcription start site.

We did indeed see an increase of coverage after the second TSS from RNA-seq data, but there are only 160 nucleotides between the first and second start sites so the resolution is not at a high enough confidence to report this data.

- Figure 3 and 6: The details of the cell culture conditions were not provided: e.g. the cells were harvested at mid-log phase? Because the level of LAerR isoform is different in each growth phase (Figure 1), the information should be given.

This information is now added to the Results section, as well as to the figure legends.

- Figure 4: The observed growth delay for dLAerR and AerR-null mutants seems to result from low-pigmentation under aerobic conditions before starting the measurement (Figure 3A).

We agree the low pigmentation under aerobic condition could indeed contributing to the lag. It is now clearly stated so in the revised Results.

- The details of the experimental conditions for the AerR and CrtJ interaction (Figure 5) were not provided. For example, it is not clear whether the AerR used in the experiment was cobalamin-bound form or unbound form. Please provide such information. If the authors used cobalamin-bound form, what kind of cobalamin bound to the apo-AerR?

The AerR-CrtJ in vitro interaction section was rewritten with the new binding results as described above. This rewriting section now addresses this issue.

- The SAerR variant, which does no longer bind cobalamin due to point mutations, still has a strong phenotype (spectra and growth). It would be appropriate to comment on this instead of claiming only slight effects.

This issue is discussed and addressed by a similar question below.

*- It is possible that the AerR point mutations influences not only cobalamin-binding characteristics, but also stability/level of AerR* in vivo*. The authors should comment on this.*

Indeed, we have checked the stability of the SAerR_H145A, G148E mutant and observed that it exhibits the same stability as does wild type SAerR (Author response image 1).

**Author response image 1. respfig1:** Western blot analysis of WT cell and cells that express flag tagged SAerR and flag tagged SAerR containing the Cbl binding mutations. Antibody used is to the flag epitope.

- LAerR, but not SAerR, expression level is changed in response to growth modes (Figure 1), and SAerR can bind both photo-activated and non-activated cobalamin although LAerR can bind only photo-activated one (Figure 7). Although the RNA seq gave some information about the functional differences of the two isoforms (Figure 9, 10), the biochemical properties of the two AerR isoforms is not clearly discussed in the manuscript. The manuscript would benefit from such discussion, with schematic models of the action of each AerR isoform and CrtJ.

We agree and have added a schematic model to the section of the Discussion that covers the repressing function of SAerR and the activating function of LAerR.

8) Some text needs to be modified:- It is sometimes hard to follow the story and to extract the important messages. The authors should make an effort for the text to be accessible to a wide audience.- It is not clear from the Abstract why photosynthetic bacteria control photosystem synthesis. What is the meaning of the first sentence of the Abstract? What do the authors mean by "adjusts their physiology for optimal light absorption"?In the first paragraph of the Introduction, the authors briefly mention that photosystems can generate oxidative stress, and that photosystem production must also be controlled to avoid oxidative stress. The important function of photoreceptors to monitor conditions, which may cause photooxidative stress and to trigger the crucial adaptation process to these conditions is not really considered in other parts of the paper.

We agree that the Abstract and Introduction were not as well written as it could have been. These sections were rewritten to improve clarity and to provide better background to a wider audience.

- The discussion of the IF3 function (Discussion, second paragraph) is inaccurate. IF3 is not 'involved in translation from non-canonical initiation codons' and it does not 'increase initiation from non-canonical codons'. Instead, IF3 discriminates against the non-canonical initiation codons.- The discussion of the hibernation factors and formation of the 100S ribosomes in the stationary phase seem irrelevant for their findings.

This section of the Discussion has been rewritten to provide the correct function of IF3.